# Optimal enzyme allocation leads to the constrained enzyme hypothesis: The Soil Enzyme Steady Allocation Model (SESAM v3.1).

Thomas Wutzler[1], Christian Reimers[1], Bernhard Ahrens[1], and Marion Schrumpf[1]

[1]Max Planck Institute for Biogeochemistry, Hans-Knöll-Straße 10, 07745 Jena, Germany

**Correspondence:** Thomas Wutzler
(twutz@bgc-jena.mpg.de)

**Abstract.** Describing the coupling of nitrogen (N), phosphorus (P), and carbon (C) cycles of land ecosystems requires understanding microbial element use efficiencies of soil organic matter (SOM) decomposition. These efficiencies are studied by the soil enzyme steady allocation model (SESAM) at decadal scale. The model assumes that the soil microbial communities and their element use efficiencies develop towards an optimum where the growth of the entire community is maximized. Specifically, SESAM approximated this growth optimization by allocating resources to several SOM degrading enzymes proportional to the revenue of these enzymes, called the Relative approach. However, a rigorous mathematical treatment of this approximation has been lacking so far.

Therefore, in this study we derive explicit formulas of enzyme allocation that maximize total return from enzymatic processing, called the Optimal approach. Further, we derive another heuristic approach that prescribes the change of allocation without the need of deriving a formulation for the optimal allocation, called the Derivative approach. When comparing predictions across these approaches, we found that the Relative approach was a special case of the Optimal approach valid at sufficiently high microbial biomass. However, at low microbial biomass, it overestimated allocation to the enzymes having lower revenues compared to the Optimal approach. The Derivative-based allocation closely tracked the Optimal allocation.

These findings increases our confidence into conclusions drawn from SESAM studies. Moreover, the new developments extend the range of conditions at which valid conclusions can be drawn. Further, based on these findings we formulated the constrained enzyme hypothesis. This hypothesis provides a complementary explanation why some substrates in soil are preserved over decades although often being decomposed within a few years in incubation experiments.

This study shows how optimality considerations lead to simplified models, new insights and new hypotheses. It is another step in deriving a simple representation of an adaptive microbial community, which is required for coupled stoichiometric CNP dynamic models that are aimed to study decadal processes beyond ecosystem scale.

## 1 Introduction

The soil enzyme steady allocation model (SESAM) studies the effect of an adaptive soil microbial community on the coupling of element cycles in aerated soils at decadal time scale. The coupling of the cycles of nitrogen (N), phosphorus (P), and

carbon (C) is especially strong in soils because the stoichiometric requirements of soil organic matter (SOM) decomposers is much less flexible than the stoichiometric requirements of plants (Robert W. Sterner, 2002; Mooshammer et al., 2014b). The stoichiometric requirements, in turn, together with the stoichiometry of consumed substrates determine decomposer's carbon and nutrient use efficiencies, which are important controls on ecosystem dynamics. Carbon use efficiency (CUE) is key to control how much of the litter input is stored in soil or respired again to the atmosphere (Manzoni et al., 2017). Similarly, nitrogen use efficiency affects how much N in litter inputs is stored in organic matter or mineralized and made available for plant nutrition (Mooshammer et al., 2014a). These element use efficiencies are also affected by properties of the microbial community. Furthermore, microbial community is hypothesized to adapt to changing environment, such as increased litter inputs or litter stoichiometry or nitrogen deposition (Manzoni, 2017; Manzoni et al., 2021).

However, there is a gap between knowledge of microbial processes at smaller and effect at larger scales. On the one hand, knowledge of the complex microbial ecology and community adaptations accumulates at the soil pore scale. On the other hand, dynamic SOM models, which rely on nutrient efficiencies of the decomposers, focus on SOM changes at ecosystem to global scale. Hence, we need to find ways to incorporate effects of soil microbial community adaptations on element use efficiencies (Kaiser et al., 2014) without the need to model all the microbial populations and microbial details.

Therefore, the SESAM model abstracts from microbial details by assuming that community composition develops towards maximizing growth of the entire microbial community. This assumption is in line with arguments from system ecology (Nielsen et al., 2020), which realized that open systems with positive internal feedback develop towards best exploiting a gradient of potential energy (Ulanowicz, 2002). This exploitation of the gradient is usually associated with maximizing entropy production that can supports internal structure of the system (Kondepudi, 1998). For soil systems this mainly translates into efficiently degrading the chemical energy input provided by plant litter and rhizodeposition. In a first approximation this efficient degradation is achieved by maximum growth and respiration of soil microbes. This focus on system perspective leads to complementary insights, compared to focusing on competition, and opens up a new ways of studying living systems (Ulanowicz, 2009). One of the problems of this argument is the question at which scale to apply the maximum entropy production hypothesis. Application at different scales leads to different predictions of optimal system dynamics (Dewar, 2010). Hence, the optimal community growth assumption is rational, but it is still an assumption to be challenged.

The heuristic approach of how community growth is optimized in SESAM requires a more rigorous treatment. The heuristics that is applied in SESAM 3.0 assumes the proportion of allocation into enzyme $Z$ to be proportional to its revenue, i.e. return per investment. Wutzler et al. (2017, Appendix B) provide a rationale of this approach, which argues that exploiting of the full range of resources is beneficial. However, this attempt does not sufficiently well explain why this leads to optimal community growth. Hence, a better, i.e more rigorous rationale is required to increase confidence into assumptions made in SESAM.

Such a rigorous treatment of the optimal enzyme allocation has now become possible because of recent model developments. The model developments of Wutzler et al. (2022) comprise a new formulation of decomposition based on quasi-steady state of enzymes and the new formulation of revenue with limitation-weighted enzyme investments. They make it possible to express the revenue directly as a function of the enzyme allocation. This functional expression now allows us, in this study, to derive

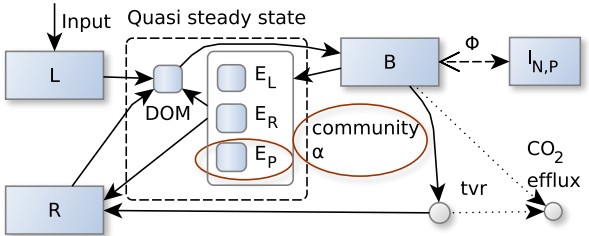

**Figure 1.** The SESAM model: Microbial biomass $B$ produces enzymes that depolymerize substrate pools ($L$ and $R$) that differ in their elemental ratios. Adaptive microbial community enzyme allocation $\boldsymbol{\alpha}$ determines which part of the microbial community depolymerizes $L$ versus $R$ by producing respective depolymerizing enzymes $E_L$, $E_R$, and a biomineralizing enzyme $E_P$ cleaving phosphate groups. Microbes take up dissolved organic matter (DOM) and use it for synthesizing new biomass, new enzymes, or for catabolic respiration. A part of microbial turnover (tvr) adds to the residue pool, another part is mineralized, and another part adds to DOM and is recycled into microbial biomass. Stoichiometric imbalance between DOM and B is resolved by mineralizing the excess element or immobilizing required element ($\Phi_B$) from inorganic N and P pools ($I$). There are additional fluxes from $L$ and $R$ to the inorganic pools, $I$, and additional plant uptake and leaching fluxes drawing from the inorganic pools, $I$, which are not shown in this figure. Boxes correspond to pools, disks to fluxes. Solid lines represent fluxes of C,N, and P, while dotted and dashed lines represent separate C,N or P fluxes respectively. Red ellipses denote changes from the Wutzler 2022 version.

optimal community allocation by maximizing the total return from enzymatic processing. Further, it inspired another simpler heuristic optimality approach.

The aim of this study is to present and compare three approaches of computing enzyme allocation, i.e. the rigorous Optimal approach, the previously applied heuristic Relative approach, and the new heuristic Derivative, approach. We compare approaches based on several scenarios of dynamic simulation and discuss the resulting insights and implications. One of those insights is the constrained enzyme hypothesis.

## 2    Methods

In this section, we first summarize the SESAM model and re-state the equations that are most relevant for the optimality approaches (subsection 2.1). Next, we present the three optimality approaches (subsection 2.2). Finally, we describe the setup of the simulation experiments (subsection 2.3).

### 2.1    The SESAM model

SESAM is described in the previous paper of this incremental model description paper series (Wutzler et al., 2022). A summary

of the model is presented by Fig. 1 and state variables, model drivers, model parameters and other symbols used in SESAM

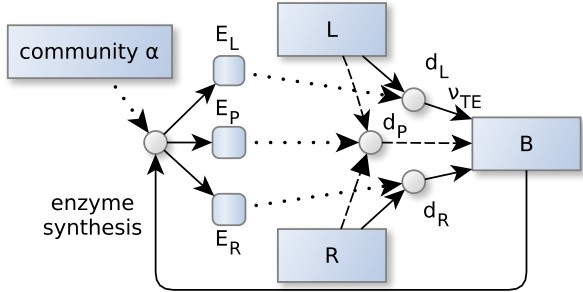

**Figure 2.** Community allocation, $\boldsymbol{\alpha}$ controls the partitioning of the enzyme synthesis. This in turn, affects the depolymerization and biomineralization fluxes of labile ($L$) and residue organic matter ($R$). $\boldsymbol{\alpha}$ adapts in such a way, so that the sum of the returns from degradation fluxes $d_Z$ is maximized. Specifically, its the part of the degradation fluxes that reaches microbial biomass $B$ by direct and indirect uptake, $\nu_{TE}$, of elements $E \in \{C, N, P\}$, and the sum is weighted by current elemental limitation of the microbes and the elemental investments required to synthesize the enzymes. Dotted lines denote controls. Other line types and shapes correspond to Fig. 1.

are listed with tables 1, 2 and 3. Symbol $d$ with a subscripts denotes a form of decomposition or return flux, while the symbol $d$ without subscripts denotes the derivative operator. The model version used in this study already anticipates ongoing unpublished model developments, which include phosphorus (P) cycling and microbial P-limitation. While P is generally handled the same way as nitrogen (N), there is an additional class of P biomineralizing enzymes, $E_P$, that does not depolymerize substrates in stoichiometric quantities, but cleaves phosphate groups. Moreover, $E_P$, is not only produced by microbes but also by plant roots. Those P related developments will be described in their own paper, but this manuscript presents formulas that can account for this new type of enzymes.

This study focuses on approximating the optimal microbial community allocation, $\boldsymbol{\alpha}$. It modifies eq. 3 in Wutzler et al. (2022) and compares several variants. SESAM substrate decomposition is controlled by the quantity of enzymes, which in turn, are controlled by the microbial community that adapts their allocation to different enzymes in order to maximize growth (Fig. 2). Allocation into different enzymes adapts to the return and revenue, i.e. return/investment, of those enzymes.

The details of how SESAM computes the return of an enzyme are restated in the following section, while revenue is described in section 2.2.1.

### 2.1.1 Depolymerizing enzymes

The return of an enzyme, $E_Z$, which depolymerizes substrate, $S_Z$, is the elemental-limitation-weighted average of the returns for the modeled elements required for microbial growth. The return equals the depolymerization flux that is taken up by microbes.

**Table 1.** State variables and model drivers. Values correspond to FACE simulation experiment initial steady state for Optimal approach.

| Symbol | Definition | Value | Unit |
|---|---|---|---|
| $L, L_N, L_P$ | C, N, and P in labile substrate | $109 \cdot \beta_{E_{i_L}}(0)$ | $\mathrm{g\,m^{-2}}$ |
| $R, R_N, R_P$ | C, N, and P in residue substrate | $3687 \cdot \beta_{E_{i_R}}(0)$ | $\mathrm{g\,m^{-2}}$ |
| $B$ | Microbial biomass C | 30.46 | $\mathrm{g\,m^{-2}}$ |
| $I_N$ | Inorganic N | 0.194 | $\mathrm{g\,m^{-2}}$ |
| $I_P$ | Inorganic P | 2157** | $\mathrm{g\,m^{-2}}$ |
| $\alpha_L, \alpha_R, \alpha_P$ | Allocation to enzyme $Z \in \{L, R, P\}$ | 0.74, 0.26, 0.0 | (-) |
| $i_L(t)$ | labile C input | 400.0 | $\mathrm{g\,m^{-2}yr^{-1}}$ |
| $\beta_{N_{i_L}}(t)$ | C:N ratio of labile inputs | 28 | $\mathrm{g\,g^{-1}}$ |
| $\beta_{N_{i_R}}(t)$ | C:N ratio of residue inputs | 10 | $\mathrm{g\,g^{-1}}$ |
| $\beta_{P_{i_L}}(t)$ | C:P ratio of labile inputs | 120 | $\mathrm{g\,g^{-1}}$ |
| $\beta_{P_{i_R}}(t)$ | C:P ratio of residue inputs | 40.3 | $\mathrm{g\,g^{-1}}$ |
| $i_{I_N}(t)$ | inorganic N input | 0.0714 | $\mathrm{g\,m^{-2}yr^{-1}}$ |
| $i_{I_P}(t)$ | inorganic P input | 0 | $\mathrm{g\,m^{-2}yr^{-1}}$ |
| $k_{I_E P}(t)$ | plant uptake of inorganic $E$ per $I_E$ | 100* | $\mathrm{yr^{-1}}$ |
| $u_{I_E,max}(t)$ | max plant uptake of $E$ | $= i_L/\beta_{E_{i_L}} + i_{I_E}$** | $\mathrm{g\,m^{-2}yr^{-1}}$ |
| $e_P$ | plant production of biomineralizing enzyme | 0 | $\mathrm{g\,m^{-2}yr^{-1}}$ |

* arbitrary high value so that plant uptake is constraint by $u_{I_E,max}(t)$

** balancing nutrient input to the system

C depolymerization flux, $d_{ZC}$ is described by reverse Michaelis-Menten kinetics (Schimel and Weintraub, 2003), which is first order to the source pool, $k_Z S_Z$, and saturating with the amount of the respective enzyme. By assuming enzymes to be near quasi-steady state at larger than month's time scale, their amount is proportional to microbial enzyme production flux ($\alpha_Z a_E B$). This enzyme production flux then is used in the Michaelis-Menten kinetics together with a lumped affinity parameter, $k_{mNZ}$.

$$d_{ZC} = k_Z S_Z \frac{\alpha_Z a_E B}{k_{mNZ} + \alpha_Z a_E B} \tag{1}$$

Nutrient, $E \in \{N, P\}$, depolymerization fluxes are derived by dividing the C flux by the C:E ratio, $\beta_{EZ}$, of the source pool. These depolymerization fluxes are then converted to C units by C:E ratio of microbial biomass, $\beta_{EB}$, so that a weighted return, $d_{Zw}$, can be computed. Further the depolymerization fluxes are multiplied by a dynamically computed proportion, $\nu_{TE}$, that describes what part of the flux currently reaches microbes rather than plants, leaching, or sequestration at minerals (Wutzler et al., 2022, Appendix B).

**Table 2.** Model parameters. Values correspond to FACE simulation experiment initial steady state for Optimal approach.

| Symbol | Definition | Value | Unit |
| --- | --- | --- | --- |
| $\beta_{NB}$ | C:N ratio of microbial biomass | 11 | $\mathrm{g\,g^{-1}}$ |
| $\beta_{NEnz}$ | C:N ratio of extracellular enzymes | 3.1 | $\mathrm{g\,g^{-1}}$ |
| $\beta_{PB}$ | C:P ratio of microbial biomass | 40 | $\mathrm{g\,g^{-1}}$ |
| $\beta_{PEnz}$ | C:P ratio of extracellular enzymes | 50 | $\mathrm{g\,g^{-1}}$ |
| $\beta_{Pm}$ | C:P ratio of a substrate at which the biomineralization decreased to 1/2 | 500 | $\mathrm{g\,g^{-1}}$ |
| $k_L$ | maximum decomposition rate of $L$ | 5.0 | $\mathrm{yr^{-1}}$ |
| $k_R$ | maximum decomposition rate of $R$ | 0.0318 | $\mathrm{yr^{-1}}$ |
| $a_E$ | enzyme production per microbial biomass | 0.365 | $\mathrm{yr^{-1}}$ |
| $k_{mN}$ | product of enzyme half saturation constant and enzyme turnover | 3.0 | $\mathrm{g\,m^{-2}\,yr^{-1}}$ |
| $\tau$ | microbial biomass turnover rate | 6.1 | $\mathrm{yr^{-1}}$ |
| $m$ | specific rate of maintenance respiration | 5.84 | $\mathrm{yr^{-1}}$ |
| $\epsilon$ | anabolic microbial C substrate efficiency | 0.68 | (-) |
| $\epsilon_{\mathrm{tvr}}$ | fraction of microbial turnover that is not mineralized | 0.3 | (-) |
| $\nu_N$ | aggregated microbial organic N use efficiency | 0.9 | (-) |
| $\nu_P$ | aggregated microbial organic P use efficiency | 0.0 | (-) |
| $i_{BN}$ | maximum microbial uptake rate of inorganic N | 0.4 | $\mathrm{yr^{-1}}$ |
| $i_{BP}$ | maximum microbial uptake rate of inorganic P | 100* | $\mathrm{yr^{-1}}$ |
| $l_N$ | inorganic N leaching rate | 0.96 | $\mathrm{yr^{-1}}$ |
| $l_P$ | inorganic P leaching rate | 0.001* | $\mathrm{yr^{-1}}$ |

* arbitrary high/low value so that system is not constrained by P

$$d_{ZN} = d_{ZC}/\beta_{NZ}$$

$$d_{ZP} = d_{ZC}/\beta_{PZ}$$

$$d_{Zw} = w_C d_{ZC}\,\nu_{TC} + w_N d_{ZN}\,\nu_{TN}\beta_{NB} + w_P d_{ZP}\,\nu_{TP}\beta_{PB}$$

$$= \frac{k_Z S_Z\,\alpha_Z a_E B}{k_{mNZ} + \alpha_Z a_E B}\left(w_C\nu_{TC} + w_N\nu_{TN}\frac{\beta_{NB}}{\beta_{NZ}} + w_P\nu_{TP}\frac{\beta_{NB}}{\beta_{PZ}}\right)$$

The limitation weighted return can be expressed as a potential return, $d_Z$, multiplied a factor that reduces return due to low enzyme levels. Hence, potential return denotes the return potentially achieved at infinitely high enzyme levels. It is the potential C-substrate depolymerization flux, multiplied by the combined elemental weighting factor $\omega_Z$.

**Table 3.** Further symbols

| Symbol | Definition | Unit |
|--------|-----------|------|
| $S \in \{L, R\}$ | Soil organic matter substrates, labile or residues | $\mathrm{g\,m^{-2}}$ |
| $Z \in \{L, R, P\}$ | Enzyme classes for depolymerizing substrates $L$ and $R$ or biomineralizing phosphorus from both substrates | $\mathrm{g\,m^{-2}}$ |
| $w_E$ | Weight of limitation of microbial growth by element $E \in \{C, N, P\}$ (eq. 4) | $-$ |
| $d_{Zw}(\alpha_Z)$ | Elemental-limitation-weighted return of enzyme $Z$ | $\mathrm{g\,m^{-2}yr^{-1}}$ |
| $d_Z$ | Elemental-limitation-weighted potential return for unlimited concentration of enzyme $Z$ | $\mathrm{g\,m^{-2}yr^{-1}}$ |
| $\omega_Z$ | Elemental-limitation factor for return of enzyme Z | $-$ |
| $\omega_{Enz}$ | Elemental-limitation factor for total enzyme synthesis in C units, $a_E B$ | $-$ |
| $u_T$ | Total return $= \sum_Z d_{Zw}(\alpha_Z)$ | $\mathrm{g\,m^{-2}yr^{-1}}$ |
| $\mathrm{rev}_Z$ | Revenue, i.e. return per investment, of enzyme Z | $\mathrm{g\,m^{-2}yr^{-1}}$ |
| $\mathrm{syn}_B$ | C for microbial biomass synthesis | $\mathrm{g\,m^{-2}yr^{-1}}$ |
| $\nu_{TE}$ | total proportions of the mineralization that are taken up by microbial biomass, $\nu_E + (1 - \nu_E)p_{immo,E}$. | $-$ |
| $\mathrm{tvr}_B$ | microbial biomass C turnover in addition to enzyme production, mostly mortality | $\mathrm{g\,m^{-2}yr^{-1}}$ |

$$d_{Zw} = d_Z \frac{\alpha_Z a_E B}{k_{mNZ} + \alpha_Z a_E B} \tag{2a}$$

$$d_Z = k_Z S_Z \omega_Z \tag{2b}$$

$$\omega_Z = w_C \nu_{TC} + w_N \nu_{TN} \frac{\beta_{NB}}{\beta_{NZ}} + w_P \nu_{TP} \frac{\beta_{PB}}{\beta_{PZ}} \tag{2c}$$

Similarly to defining an elemental-weighted limitation factor for enzyme returns, such an elemental-weighted factor is defined for enzyme synthesis flux (3).

$$\omega_{Enz} = w_C + w_N \frac{\beta_{NB}}{\beta_{NEnz}} + w_P \frac{\beta_{PB}}{\beta_{PEnz}} \tag{3}$$

The elemental weights, $w_E$ are the same in both, $\omega_Z$ and $\omega_{Enz}$. Therefore, they do not need to be normalized in ratios of these two quantities, e.g. the revenue calculation in section (2.2.1).

How strongly microbial biomass is limited by either of the elements $E \in \{C, N, P\}$, is described by the elemental limitation weights (4) (Wutzler et al., 2022, A15).

$$w_E = \exp\left(-\delta \frac{C_{\mathrm{synBE}} - \mathrm{syn}_B}{\mathrm{tvr}_B}\right), \tag{4}$$

It exponentially decreases with the difference between flux potentially available for microbial biomass synthesis by this element, $C_{\mathrm{synBE}}$, and the actual synthesis flux, $\mathrm{syn}_B$, which is constrained also by other elements. To derive a unitless quantity,

it is scaled by the microbial turnover flux, $\mathrm{tvr}_B$. Parameter $\delta$ controls, how steep is the transition near co-limitation by several elements. For an more detailed presentation of the elemental limitation we refer the reader to Wutzler et al. (2022).

### 2.1.2 Biomineralizing enzymes

The phosphatases only cleave phosphate groups from soil organic matter. Hence, they make available only P for uptake,
without making available C and N. They attack both labile and residue organic matter. Although the P-cycle in SESAM will be described in its own manuscript, here, we state the return and revenue.

The potential return of action of P-degrading enzymes, $d_P$, includes the P-limitation weights $w_P$ only, contrary to the depolymerizing enzymes (2), Moreover, it does not divide by the C:P ratio of the substrate, as the mineralization flux is already expressed in P units:

$$d_{Pw} = d_P \frac{\alpha_P a_E B}{k_{mNP} + \alpha_P a_E B} \tag{5a}$$

$$d_P = \omega_P (k_{LP} l_{\beta_{P_L}} L_P + k_{RP} l_{\beta_{P_R}} R_P) \tag{5b}$$

$$\omega_P = w_P \nu_P \beta_{PB} \tag{5c}$$

$$l_{\beta_{PS}} = \frac{1}{1 + \beta_{PS}/\beta_{Pm}} = \frac{\beta_{Pm}}{\beta_{Pm} + \beta_{PS}} \tag{5d}$$

In addition, a limitation factor $l_{\beta_{PS}} \in (0,1)$ decreases the potential rate of a biomineralizing enzyme with increasing C:P ratio,
$\beta_{PS}$, of substrate $S$. Parameter $\beta_{Pm}$ is the C:P ratio at which the limitation factor decreased to 1/2.

Moreover, these phosphatases are also produced by plant roots at a rate $e_P$. Hence, one needs to calculate the return of microbe-produced enzymes, $d_{Pm}$, by subtracting the flux due to plant-produced enzymes, from total biomineralization flux (Table B1).

## 2.2 Allocation optimization approaches

The derivative of the total return, $u_T$, with respect to each enzyme allocation share, $\alpha_Z$, for short called 'the derivative' is the central quantity to inspect. The differences across those derivatives across enzymes determine the direction of changes in enzyme allocation, i.e. changes in microbial community. Allocation is changed towards the enzyme $Z$ with the highest derivative, i.e. highest increase in return per additional allocation, at the expense of decreasing allocation to enzymes with the lowest derivative. Hence, derivatives are equal at the optimum (Appendix B1). The derivatives decrease with increasing
allocation because the return saturates at high enzyme levels. Therefore, it is often beneficial for the community to distribute investment into enzymes across different enzymes rather than investing solely into the enzyme with the highest potential return (Fig. 3).

The revenue of allocation to enzyme $Z$, another important quantity, is the return from enzymatic processing (sections 2.1.1 and 2.1.2) divided by the investment into enzyme production: $u_Z = \frac{d_{Zw}(\alpha_Z)}{\alpha_Z \omega_{Enz} a_E B}$. The investment is the share, $\alpha_Z$, invested
into production of enzyme $Z$, multiplied by total elemental-limitation-weighted carbon flux allocated to enzyme production, $\omega_{Enz} a_E B$.

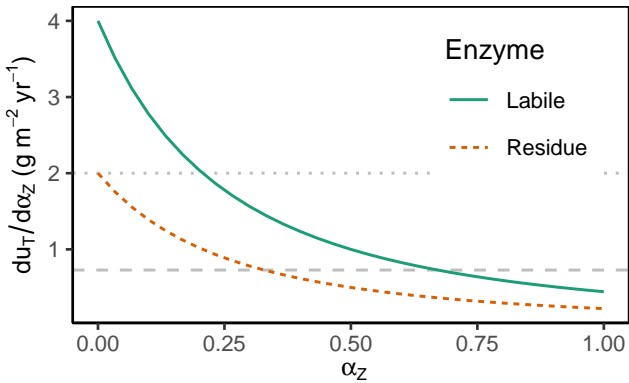

**Figure 3.** The derivative of total return with respect to enzyme allocation, $\frac{du_T}{d\alpha_Z}$, decreases with increasing share of allocation $\alpha_Z$. Therefore, when going from zero allocation proportions ($\alpha_L = \alpha_R = 0$) towards complete allocation ($\alpha_L + \alpha_R = 1$), in the shown example, microbes first increase allocation to enzymes that degrade labile substrates, $\alpha_L$, which yields the highest positive change in return. However, starting at levels $\alpha_L > 0.25$ (indicated by the dotted horizontal line crossing the solid "Labile" derivative line), the increase in return with only increasing $\alpha_L$ is less or equal to the increase in return when also allocating something to residue degrading enzymes, $\alpha_R > 0$. Optimal allocation is attained when both derivatives are equal and allocation proportions add up to one (indicated by dashed horizontal line). This happens here at allocation about 1/3 to residue depolymerizing enzymes ($\alpha_R = 1/3$) and 2/3 to labile pool depolymerizing enzymes ($\alpha_L = 2/3$).

Hence total return and revenue depend on the potential decomposition flux, i.e. the amount and the decomposition rate of the substrate, as well as its stoichiometry via weighting by current elemental limitation of the microbes. In addition, they depend on enzyme levels, i.e. the size of the microbial biomass producing the enzymes, and on the current enzyme allocation, i.e. the

shares of total enzyme production into the alternative enzymes.

Three approaches of estimating the time development of enzyme allocation, $\boldsymbol{\alpha}$ are presented in this study. The Optimal approach is the mathematically exact formulation of the hypothesis of maximum return of enzyme investment, but is only practical for simple cases. Therefore, two heuristic approximations are added. First, the Relative approach assumes that the optimal allocation can be estimated by setting the allocation proportional to the revenue. Second, the Derivative approach

describes the direction of change in allocation without explicitly computing the optimal allocation.

The optimum, to which microbial community in SESAM develops towards, is characterized by maximum growth of the entire microbial community, which in turn is achieved when the return from extracellular enzymatic processing is maximized.

### 2.2.1 Total return of enzyme action

We seek the enzyme allocation $\boldsymbol{\alpha}$ that maximizes the total limitation-weighted return, i.e. the action of enzymes, depolymeriza-

tion and biomineralization. We exclude the trivial case of investing only into a single enzyme, ($\alpha_Z = 1$), and exclude enzymes that are not allocated to ($\alpha_Z = 0$).

The total return that is optimized is the the sum of each revenue multiplied by enzyme investment.

$$u_T = \omega_{Enz} a_E B \sum_Z \alpha_Z \operatorname{rev}_Z \tag{6}$$

$u_T$ fulfills the conditions of Lemma 1 (Appendix B1). Therefore, potential optima are located at the borders or at condition $\frac{d(\alpha_Z \operatorname{rev}_Z)}{d\alpha_Z} = C_3$. This implies that the derivatives of total return, $\frac{du_T}{d\alpha_Z} = \omega_{Enz} a_E B \frac{d(\alpha_Z \operatorname{rev}_Z)}{d\alpha_Z}$, are equal at the optimum.

The revenue for a depolymerizing enzyme and its derivative are

$$
\begin{aligned}
\operatorname{rev}_Z(\alpha_Z) &= \frac{\text{return}}{\text{investment}} = \frac{d_{Zw}}{\alpha_Z \omega_{Enz} a_E B} \\
&= d_Z \frac{\alpha_Z a_E B}{k_{mNZ} + \alpha_Z a_E B} \frac{1}{\alpha_Z \omega_{Enz} a_E B} \\
&= \frac{d_Z}{\omega_{Enz}} \frac{1}{k_{mNZ} + \alpha_Z a_E B}
\end{aligned} \tag{7a}
$$

$$
\begin{aligned}
\frac{d(\alpha_Z \operatorname{rev}_Z)}{d\alpha_Z} &= \frac{d_Z}{\omega_{Enz}} \frac{(k_{mNZ} + \alpha_Z a_E B) - \alpha_Z a_E B}{(k_{mNZ} + \alpha_Z a_E B)^2} \\
&= \frac{d_Z}{\omega_{Enz}} \frac{k_{mNZ}}{(k_{mNZ} + \alpha_Z a_E B)^2}
\end{aligned} \tag{7b}
$$

Revenue $\operatorname{rev}_P$ and its derivative of a biomineralizing enzyme are slightly more complex due to plant enzyme production but, they are similar to the ones of the depolymerizing enzymes. They are presented by appendix Table B1.

### 2.2.2 Optimal approach

The Optimal approach computes the target allocation that maximizes total return by computing where the derivatives of total return across the set of allocated enzymes are equal (Appendix B1). Such a derivative of the return with respect to enzyme allocation $\alpha_Z$ is proportional to the derivative of the allocation times the revenue, $\frac{du_T}{d\alpha_Z} \propto \frac{d(\alpha_Z \operatorname{rev}_Z)}{d\alpha_Z}$ (section 2.2.1). While the maximum change of return is realized at an arbitrarily small allocation $\rho_{Zmax} = \frac{d(\alpha_Z \operatorname{rev}_Z)}{d\alpha_Z}\big|_{\alpha_Z \to 0}$, the optimal allocation $\boldsymbol{\alpha}^*$ often involves several enzymes (Fig. 3). However, if the maximum change of return for an enzyme $Z_j$ is lower than the return of allocating only to other enzymes, the optimal allocation to this enzyme is zero, i.e. it is excluded from the set of allocated enzymes. The set of allocated enzymes, i.e. enzymes among which to distribute resources, can be found by the following algorithm.

1. Order the enzymes according to their maximum change in return, $\rho_{Zmax}$, index them by $i$, set $i = 1$ and start with a mix that includes only the most efficient enzyme $\{Z_1\}$.

2. Solve for the optimal allocation strategy $\boldsymbol{\alpha}_i$ equalizing derivatives:

$$\frac{du_T}{d\alpha_Z} \propto \frac{d(\alpha_Z \operatorname{rev}_Z)}{d\alpha_Z} = \rho_i \text{ for all } Z \in \{Z_1, \dots, Z_i\}$$

and allocate nothing to enzymes that are not part of the current mix.

3. For $\rho_i$ computed in step 2, if $\rho_i > \rho_{Z_{i+1}max}$ stop and report the found optimum $\boldsymbol{\alpha}^* = \boldsymbol{\alpha}_i$. Otherwise increase $i$, i.e. include enzyme $Z_{i+1}$ in the mix and go to step 2.

Step 2 needs explicit solutions for different numbers and types of enzymes in the mix. Appendix B3 provides such explicit solutions for up to three enzymes across depolymerizing and biomineralizing enzymes.

### 2.2.3 Relative approach

The Relative approach, which was used up to SESAM version 3.0 (Wutzler et al., 2022), estimates optimal allocation to be proportional to revenue based on current allocation (8).

$$\alpha_{Z,Opt} = \frac{\text{rev}_Z}{\sum_i \text{rev}_i} \tag{8}$$

where $\text{rev}_Z$ is the revenue for enzyme $Z$.

Appendix C shows that it is a special case of the Optimal approach given several assumptions. It well approximates optimal allocation for the case of sufficiently high microbial biomass levels.

### 2.2.4 Derivative approach

The Derivative approach computes the rate change of $\alpha_Z$ over time. It assumes that enzymes allocation changes faster, the larger the corresponding derivative is away from the average, i.e. the optimal state where all derivatives are equal. More precisely, it assumes the change rate of allocation over time to be $\frac{d\alpha_Z}{dt} \propto \frac{du_T}{d\alpha_Z} - \text{mean}_i\left(\frac{du_T}{d\alpha_i}\right)$ across the enzymes in the current mix (Appendix D). It does not rely on an optimal solution $\boldsymbol{\alpha}^*$. This is beneficial, because formulas in the Optimal approach for a higher number of enzymes or more types of enzymes quickly grow and involve higher-order polynomials of $\alpha_Z$ with multiple roots and additional mathematically possible solutions outside the reasonable bound $\alpha_Z \in [0,1]$.

The Derivative approach assumes that higher increase in total return lead to faster shifts of allocation towards this enzyme. It takes care, similar to the Optimal approach, to compute the average only across enzymes that are part of the current mix (Appendix D1).

### 2.3 Simulation experiments

In order to study the effects of using different allocation optimization approaches on model behavior, we set up different simulation experiments and compared differences in predictions among the approaches.

### 2.3.1 Immediate response: Prescribed potential returns

The Prescribed potential returns simulation experiment fixed the direct inputs to the function computing allocation changes. It neglected all other model feedback and focused and compared computation of optimum allocation for prescribed conditions.

Specifically, the experiment prescribed elemental-limitation-weighted potential return fluxes, $d_Z$ (section 2.1.1), which otherwise had been dynamically computed in the model from pools and parameters. It assigned values for enzymes decomposing residue litter and biomineralizing phosphorus of $d_R = 0.7\text{gC}\,\text{m}^{-2}\,\text{yr}^{-1}$, $d_P = 0.5\text{gC}\,\text{m}^{-2}\,\text{yr}^{-1}$, and varied the flux for enzymes decomposing labile substrates $d_L \in \{0\ldots1\}$ in units $\text{gC}\,\text{m}^{-2}\,\text{yr}^{-1}$. It simulated the allocation state until it converged

to its estimated optimum for each $d_L$. For complete reference we list the other relevant parameters without further explanation here: $a_E = 0.1 \mathrm{yr}^{-1}, B = 1 \mathrm{gC\,m}^{-2}, e_P = 0 \mathrm{gC\,m}^{-2}\,\mathrm{yr}^{-1}, \tau = 365/30 \mathrm{yr}^{-1}, k_{mN} = a_E B/2, \omega_{Enz} = 1$. The experiment included further runs with five-fold increased microbial biomass levels, $B$.

### 2.3.2 Decadal-term: FACE

The FACE simulation experiment simulated the decadal-term response of the system to increased labile substrate inputs. It started with model pools in steady-state with litter inputs. Next it prescribed a jump of labile substrate inputs by 20% simulated for 50 years and prescribed another jump of labile substrate inputs to initial values. It simulated N-limited conditions and excluded P-limitation by prescribing an arbitrary high value of potential P immobilization and very low P leaching (Table 2). The experiment included two additional scenarios where parameters with the Relative approach had been adjusted to match the initial steady-state conditions of the Optimal approach. These additional scenarios allowed testing if the differences in predictions could be compensated by other model parameters.

### 2.3.3 Sub-annual: Incubation

The Incubation simulation experiment added a portion of labile substrate to a previously labile-substrate-depleted soil. Next, it tracked the carbon use efficiency (CUE) of the microbial community over time and across different C:N ratios of the added labile substrate. Specifically, it first simulated model pools in steady state with continuous annual inputs, then simulated no inputs for one year in order to deplete the labile substrate pool, and next it simulated a step-increase of the labile substrate C and N pools. In addition to the three scenarios that differed by optimality approach, it simulated a scenario where microbial community allocation was fixed to $\alpha_R = 0.5$. This scenario allowed comparing results to a model where microbial community is not adaptive.

We do not expect simulating a correct time-dynamics with SESAM at this short time scale, because SESAM explicitly omits detailed microbial processes that are relevant at this scale such as storage, resting stages, or dynamics of the enzyme pools. However, the experiment allows inspecting general dynamics with smooth annual changes and differences between model variants as the labile substrate pool gets depleted.

## 3 Results

In this section, we present the results of the simulation experiments in turn.

### 3.1 Prescribed potential returns experiment

The Derivative approach yielded the same allocation as the Optimal approach with the Prescribed potential returns simulation experiment. The Relative approach yielded similar results as the Optimal approach for high microbial biomass levels, i.e. levels that resulted in an enzyme synthesis flux of 10 times the half-saturation constant of enzyme action $k_{mN}$, which in SESAM is a flux, specifically the product of a half-saturation enzyme concentration and enzyme turnover rate. For moderate

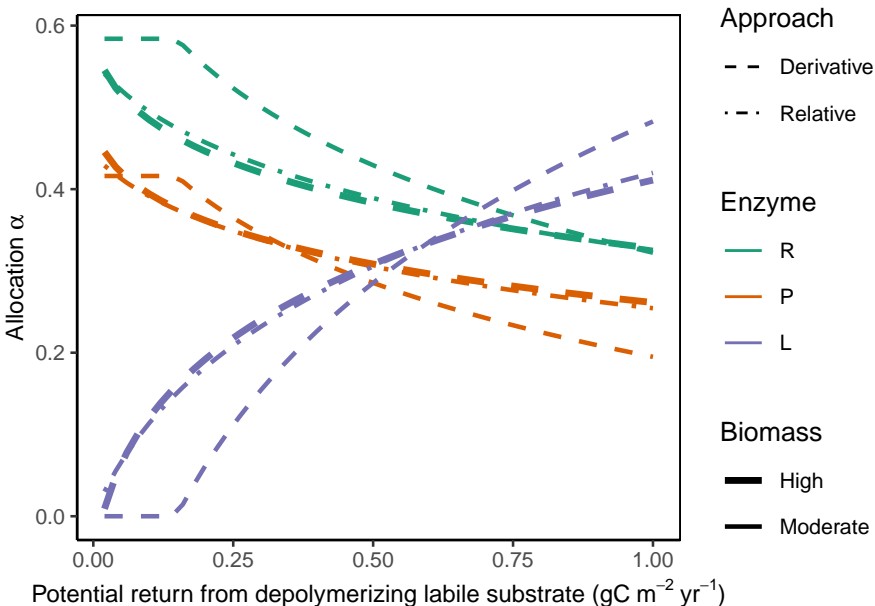

**Figure 4.** In the Prescribed potential returns simulation experiment, all allocation approaches predicted the same pattern of increasing allocation to the enzymes degrading the labile substrate, $\alpha_L$ with increasing potential return from depolymerizing labile substrate and a corresponding decrease of allocation to the other enzymes, $\alpha_R$ (residue depolymerizing)) and $\alpha_P$ (phophorus biomineralizing) respectively. The Derivative approach (dashed lines) and the Optimal approach (same predictions as Derivative, not shown) allocated nothing to the L depolymerizing enzyme at low potential returns at moderate microbial biomass levels. The Relative approach (dash-dotted lines) predicted very similar allocation as the Derivative approach at higher microbial biomass levels (indicated by overplotting of the thick lines), but overestimated allocation to enzymes of low revenue at moderate biomass levels (thin lines).

microbial biomass levels it overestimated allocation to the enzymes with low revenue (Fig. 4). With the Optimal and Derivative approaches there was no investment into enzymes with very low revenue at moderate biomass levels.

Since all state variables are held constant in this experiment, there is no change in respiration, microbial growth, CUE, as well as C in key model compartments. These subsequent changes are explored in the following experiments.

### 3.2 Decadal-term: FACE

The Derivative approach yielded the same allocation as the Optimal approach with the FACE simulation experiment. The Relative approach differed by overestimating the allocation to the enzyme with lowest revenue, $\alpha_R$. Hence, it predicted smaller initial steady state stocks but also predicted relatively less mining of residue OM during period of increased carbon inputs (Fig. 5). By adjusting parameters related to organic matter decomposition in the simulation with the Relative approach, the same steady state stocks were simulated, but still the decrease of residue OM was smaller (Fig. A1).

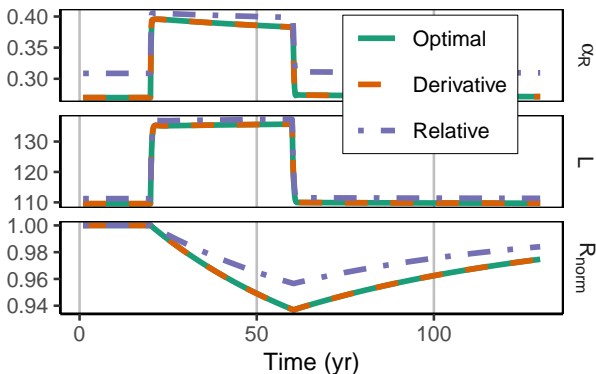

**Figure 5.** In the FACE simulation experiment all three allocation approaches predicted the same pattern of increased labile substrate ($L$ in $\mathrm{gCm}^{-2}$) and a shift towards mineralization of residue substrate (R normalized by initial steady state values). The Derivative approach yielded the same predictions as the Optimal approach (indicated by dashed line overplotting the solid line). The Relative approach (dash-dotted line) slightly overestimated allocation to the residue degrading enzymes, $\alpha_R$. This resulted in lower initial R stocks and a smaller decrease in the period of higher carbon inputs between year 10 and 60.

### 3.3 Sub-annual: Incubation

The difference between optimization approaches were small compared to the differences to the variant without adaptation (Fig. 6). All three optimization approaches showed decreased fluctuations of CUE, both in time, as well as across C:N ratios of added labile substrate compared to a non-optimized fixed allocation. The Derivative approach's predictions matched the Optimal approach's predictions, while the Relative approach initially slightly underestimated allocation to the residue degrading enzyme ($\alpha_R$) resulting in decreased biomass synthesis (Fig. 7).

The Relative approach's predictions differed from Optimal approach after one year of incubation when microbial biomass and enzyme levels declined (Fig. 8). It still allocated to the enzymes degrading labile substrate ($\alpha_R < 1$), while with the Optimal approach, microbial community did not invest into degrading the small labile substrate pool anymore. Hence, some of the labile substrate pool was not decomposed, i.e. was apparently persistent with the Optimal approach.

### 4 Discussion

The purpose of this work was to more rigorously define and implement the optimal growth hypothesis for SESAM and study the consequences of two simplifications. We found that the previously used Revenue approach could be derived from the more rigorous Optimal approach for a set of conditions. Therefore, we are more confident into conclusions drawn from previous SESAM studies. Further, we found no or only marginal differences between the Derivative and Optimal approaches. There-

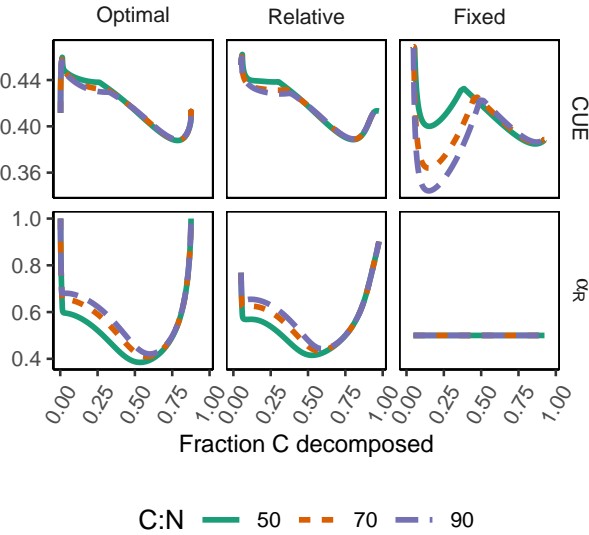

**Figure 6.** The differences in predicted carbon use efficiency (CUE) were small across optimization approaches (first two panels) compared to non-adaptive Fixed allocation in the Incubation simulation experiment. Differences in allocation to residue degrading enzymes, $\alpha_R(\mathrm{gg}^{-1})$, are constrained to the very start and end of the experiment.

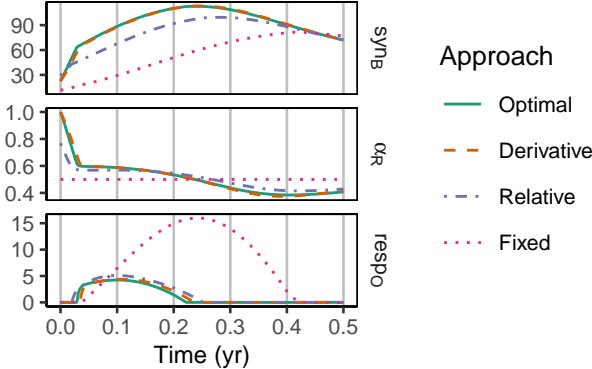

**Figure 7.** In the Sub-annual incubation simulation experiment the three optimization approaches yielded a higher biomass synthesis, $\mathrm{syn}_B$ $(\mathrm{gCm}^{-2}\mathrm{yr}^{-1})$, than Fixed, i.e. not adapting allocation. They allocated relatively more resources to the residue degrading enzymes $\alpha_R$ during the initial N-limitation. This resulted in lower overflow respiration, $\mathrm{resp}_O$ $(\mathrm{gCm}^{-2}\mathrm{yr}^{-1})$. The Relative approach initially underestimated $\alpha_R$ resulting in slightly lower biomass synthesis compared to the Optimal approach. Shown predictions correspond to an amendment with C:N ratio of 50 g/g.

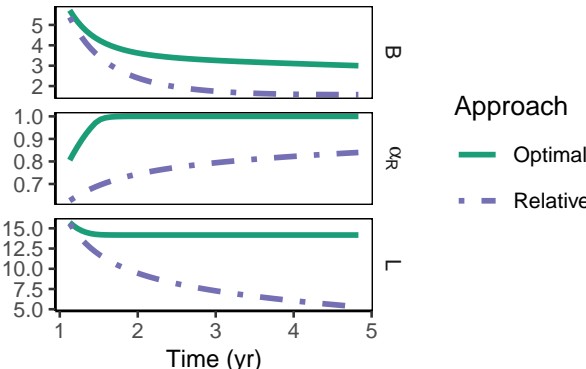

**Figure 8.** In the Incubation simulation experiment after some time microbial biomass, $B$ (gCm$^{-2}$) decreased to low levels and allocation shifted towards residue degrading enzymes only, $\alpha_R = 1$ with the Optimal approach (solid line). Hence, decomposition of a small remaining pool of labile substrate, $L$ (gCm$^{-2}$), stopped.

fore, we will use the Derivative approach to further develop the SESAM. The following section discusses the optimization approaches in more detail.

### 4.1 Optimization approaches

The Optimal approach constitutes the mathematical formalization of the hypothesis of community enzyme allocation optimizing microbial biomass growth for SESAM. The Relative approach has been used in previous SESAM versions. It has been shown in this study to be a special case of the mathematically formalized Optimal approach. It is valid for enzyme allocation fluxes larger than the half-saturation constant in the decomposition equation, which is usually the case at not too small microbial biomass. The Derivative approach is another heuristic of optimal enzyme allocation that relies on derivatives of the enzyme returns but does not require explicit formulas for the optimal allocation.

The three approaches predicted the same patterns in long-term as well as sub-annual scale simulation experiments. Hence, the conclusions drawn with SESAM so far were corroborated. Specifically, the following patterns emerge as a consequence of microbial community adaptation of enzyme allocation: The priming effect (Kuzyakov et al., 2000) and the N banking mechanism (Perveen et al., 2014), (Fig. 5), and the dampening of CUE fluctuations with an adaptive microbial biomass (Kaiser et al., 2014) (Fig. 6).

While the Optimal approach is exact, it is tedious to implement and to update with further developments of SESAM. It requires the developers to derive explicit formulas for the optimal allocation for each combination of enzyme types in the mix of enzymes allocated to. With including more enzymes or more types of enzymes, the formulas grow complex and an increasing number of potential optima need to be checked and compared. Therefore, we also consider the simpler Relative and Derivative approaches and discuss their effect on model predictions.

The Derivative approach yielded predictions that were so close to the predictions of the Optimal approach that they can hardly be spotted in the plots (Figs. 5, 7). However, it works similar to gradient based numerical optimization schemes and also shares its risks. First, it might result in limit cycles, where residue organic matter and microbial biomass oscillate instead of converge to a stable optimal allocation. We argue that this actually may really happen in soil, although perturbations with fluctuating litter input and decomposition fluxes changing with environmental conditions may quickly shift the decomposer system into states away from the basin of such a limit cycle (Strogatz, 1994). If the Derivative approach yields predictions with a decadal-scale limit cycle, perturbations of model drivers are expected to drive the simulation away from the limit cycle. Second, the Derivative approach might get stuck in local optima and saddle points where the derivative of the return approaches zero. Gradient based optimization schemes implement some notion of momentum to get past such points. There is also some momentum in the soil system, because enzyme levels need some time to develop towards its quasi-steady state and microbes use storage compounds to support developments in hourly to weekly time scale where returns from enzymatic processing do not support further growth. Because SESAM explicitly tries to abstract from such microbial details that are important for reacting on short-term fluctuations, the Derivative approach is prone to this risk of getting stuck at saddle points. We did not encounter such conditions at our simulations yet. However, in case such issues pose a problem, we need to think of ways how to implement simple notions of momentum in SESAM.

The Relative approach yielded predictions that differed from the predictions of the Optimal approach, specifically for low microbial biomass levels and for enzymes with low revenue. This was expected with the derivation of the conditions where the Relative approach is valid (Appendix C). Although small differences in enzyme allocation yield also only small differences in relative steady-state stocks, a small relative difference in the stock of the residue pool can result in considerable differences of total soil organic matter stocks. Such behaviour is observed in the FACE simulation experiments (Fig. 5). With this experiment, the Relative approach predicted an initial share of enzyme allocation towards residue degrading enzymes of 30% compared to about 26% with the Optimal approach. This led to a decrease of residue steady state stocks from about 3600 to about 3400 $gCm^{-2}$ (Fig. A1), which is an absolute difference that was larger than the entire labile substrate pool. This, in turn, resulted in a predicted relative change of residue stocks with the FACE simulation experiment that significantly differed from the predictions with the Optimal approach (Fig. 5)

Based on these findings, we will continue developing SESAM focusing on the Derivative approach.

## 4.2 The constrained enzyme hypothesis

The Optimal approach's predictions differed most from the previously used Relative approach's predictions at low microbial biomass levels. The Optimal approach excluded enzymes with low revenue from the set of enzymes to allocate to. For example, the allocation to the enzyme depolymerizing the labile substrate pool was zero for a potential return of this enzyme below 0.2 $g\,m^{-2}yr^{-1}$ in the Prescribed potential returns simulation (Fig. 4). The optimal enzyme allocation is determined primarily by availability of carbon and nutrients from organic and inorganic uptake. However, with the Optimal approach, the optimal enzyme allocation in addition depends on the size of the microbial biomass, because they control the relative size of the enzyme pools compared to saturating levels. The lower the microbial biomass, the farther away is enzyme production from

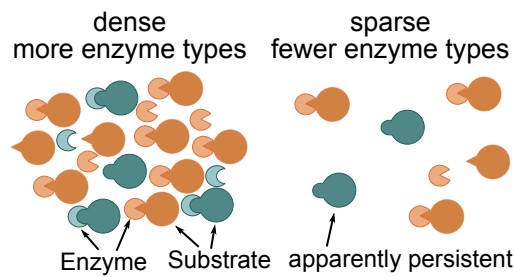

**Figure 9.** Constrained enzyme hypothesis: At low microbial concentrations, it is not beneficial for the microbial community to allocate to different enzymes types. There are some substrates for which no enzymes are synthesized. Hence, some substrates, which may be quickly decomposed at higher microbial concentrations, become apparently persistent. This apparent persistence only indirectly depends on the properties and accessibility of the substrates and depends more on the relative availability of alternative substrates.

levels where organic matter decomposition saturates. Hence at low microbial biomass it is not beneficial to distribute enzyme allocation across several enzymes including enzymes with low potential revenue. Similarly, the Optimal approach predicted in the Incubation simulation experiment that a small fraction of added organic matter, $L$ is not decomposed (Fig. 8). This insight into optimal allocation with SESAM generates an additional hypothesis why we observe high ages of some organic matter in soil and an additional insight into priming mechanisms (Fig. 9): Microbial community expresses a smaller set of enzyme types at low biomass levels. This hypothesis predicts that some organic matter is not decomposed in the presence of microbes that potentially can decompose it, if biomass levels are low and if there are alternative substrates decomposable with higher revenue.

The constrained enzyme hypothesis is able to account for observed rhizosphere priming (Cheng et al., 2014) or increased SOM loss after disturbance. When microbial biomass grows, e.g. by making enough labile substrate available, the focus on solely the enzymes with highest revenue is not beneficial any more and the optimal microbial community also invests into decomposition of the organic matter with lower revenue. Apparently persistent organic matter becomes decomposed.

This non-investment into enzymes of relatively low revenue is a complementary hypothesis to the existing hypotheses of SOM preservation (von Lützow et al., 2006, 2008). After the chemical recalcitrance hypothesis has largely been dismissed (Schmidt et al., 2011; Dungait et al., 2012; Kögel-Knabner, 2017) most hypotheses focus on restricted accessibility of SOM to soil microbial decomposition. One set of hypotheses emphasizes protection by association with minerals (Schrumpf et al., 2013; Ahrens et al., 2015; Mathieu et al., 2015; Woolf and Lehmann, 2019), another set emphasizes protection inside soil aggregates (Six et al., 2000; Lehmann, 2007; Schrumpf et al., 2013), another set emphasizes soil heterogeneity and spatial separation (Ekschmitt et al., 2005, 2008; Salome et al., 2009) or environmental conditions (Or et al., 2007; Keiluweit et al., 2017). They are related to the hypothesis of microbial energy limitation and are modified by inputs of fresh organic matter, i.e. the priming effect (Keiluweit et al., 2015; Henneron et al., 2022). Recently, the diversity hypothesis has gained attention for SOM preservation (Lehmann et al., 2020; Weverka et al., 2023), which has formerly been discussed in aquatic literature (Jannasch,

1967; Jiao et al., 2010; Arrieta et al., 2015). It emphasizes the low return on investment for very heterogeneous substrates and the associated required investment into a broad set of enzymes. The constrained enzyme hypothesis goes beyond the diversity hypothesis. While in the diversity hypothesis, the preservation is controlled by the heterogeneity of available substrates, the constrained enzyme hypothesis predicts that preservation additionally depends on the amount or density of microbial biomass and on the availability of alternative substrates.

Very similar conclusions have been drawn by a modeling study that was published shortly after the discussion paper of this study. Weverka et al. (2023) modeled the revenue of intracellular enzymes or metabolic pathways that need to be expressed to assimilate a diverse set of substrates. Similar to Wutzler et al. (2017), they compared different strategies of microbes investing into different enzymes including a strategy of investing only into the enzyme with highest revenue and a strategy the corresponds to the relative/revenue approach. They also assumed microbes to maximize growth. Their model structures, formulas for allocation and insights are comparable to this study. They differ by focusing on assimilation and intracellular enzymes, rather than decomposition and extracellular enzymes, and they focus on the number of different substrates rather than stoichiometry of substrates. Instead of computing optimal allocation they assumed that microbes would not invest into enzymes where change in return was less than investments (Harvey et al., 2016).

Similar to this study, they observed in their model that substrates at low concentration persist because it is not beneficial for microbes to produce respective enzymes. Moreover, they explained cometabolization of substrate of low revenue by the assumptions that assimilation/degradation saturates at high enzyme levels and that is therefore beneficial for microbes to distribute their investments also into enzymes of lower potential return (Fig. 3). The current study differs from Weverka et al. (2023) by actually computing the optimal enzyme allocation, and consequently predicts different allocation and conditions, at which specific enzymes are produced.

### 4.3 Optimality assumptions

The conclusions of this paper depend on several assumptions. First, they depend on the formulation of depolymerization (1) and biomineralization (5). Specifically, they depend on the assumption that the decomposition fluxes saturate at high enzyme levels (Schimel and Weintraub, 2003; Tang and Riley, 2019). With alternative formulations (Wutzler and Reichstein, 2008) that assume a linear dependence of decomposition on enzyme levels (or alternatively microbial biomass) it would be optimal to allocate to the single enzyme that yields the highest decomposition flux of the currently limiting element.

Moreover, we assumed that the instantaneous growth rate of the microbial community is optimized. Alternatively, to instantaneous growth, the cumulative growth over a microbial characteristic time-span could be optimized, e.g. the time for decomposing a single portion of carbon (Manzoni et al., 2023). The instantaneous strategy is sub-optimal to dynamical strategies if legacy effects are present that are internal to the optimized system. At the same time the two strategies yield similar performance when legacy effects are external to the optimized system (Feng et al., 2022), because competition alters the trade-off between current and future gains. Hence, optimizing at a different system boundary, which is usually associated with a different time scale, results in different optimal strategies (Dewar, 2010). The focus of SESAM on the entire microbial community calls for a dynamic strategy because it renders many factors internal, compared to a focus on competing microbial

populations that renders soil organic matter an external factor. However, SESAM is intended to model decadal-term changes and to be driven with annually averaged drivers. The two strategies will presumably converge at such conditions. This is because enzyme pools and decomposition develop towards a quasi-steady state where current and future gains are similar within a sub-annual timescale of microbial growth.

SESAM focused on the partitioning of allocation of the total enzyme investment towards different enzymes. In addition, the total allocation into enzyme production can be a trait that adapts to optimize microbial growth (Calabrese et al., 2022). Future SESAM developments will explore if a joint optimization of total allocation and allocation partitioning can be derived, and whether such a joint optimization alters the consequences for the long-term dynamics of SOM stocks.

### 4.4   Observational evidendence

The constrained enzyme hypothesis is a consequence of several model assumptions. It was derived without reference to observed patterns. However, there is already some observational evidence supporting the hypothesis of lower diversity of expressed enzymes at low microbial activity.

    Metatranscriptomics (Carvalhais et al., 2012) directly studies functional diversity of expressed enzymes in soils. Evidence for the constrained enzyme hypothesis resulting from such studies are mixed. Straw amendmends increased microbial activity

diversity of an agricultural soil and let microbes upregulate several enzyme families (Kozjek et al., 2023). This result is in line with the constrained enzyme hypothesis. Contrary, microbes downregulated enzyme families with straw amendmend to a soil of an already diverse grassland soil in the same study.

    A novel approach combines isotopically labeled measurements of microbial growth with quantitative stable isotope probing (Hungate et al., 2015). It can assess microbial diversity of the active part of the microbial community. It revealed a reduction of

diversity of actively growing microorganisms with lower microbial activity under drought (Richter, 2023), which is in line with expected reduction in diversity of expressed enzymes with lower microbial biomass as predicted with the constrained enzyme hypothesis. However, low diversity of actively growing microorganisms under drought could also be due to stress-induced shifts toward non-active conditions rather than due to optimal allocation with lower active microbial biomass.

    Analysis of potential activities of specific enzymes (Marx et al., 2001) and its spatially resolved zymography version (Spohn

et al., 2013) do not directly tell about the diversity of enzyme expression, because only specific enzymes are analyzed. However, in line with the constrained hypothesis, zymography of a temperate forest soil revealed that common enzymes are hardly expressed outside hotspots and before fostering microbial growth by amendments (Heitkötter and Marschner, 2018).

    In summary, studies that specifically look at enzyme diversity in relation to microbial biomass levels are still lacking. However, we can find observations from other studies that are in line with the constrained enzyme hypothesis.

## 5   Conclusions

The Optimal approach is the mathematical formulation of the hypothesis that microbial community enzyme allocation develops in a way that optimizes growth in SESAM. The finding of similar predictions by the heuristic approaches compared to the

Optimal approach increases our confidence into conclusions drawn with SESAM. The heuristic Relative approach is shown to be a special case of the Optimal approach valid at sufficiently high microbial biomass levels. The Derivative approach, another heuristic of the Optimal approach, is valid also for low microbial biomass levels. Given that the Derivative approach is a good heuristic of the Optimal approach that is better scalable to more enzyme types than the Optimal approach, we will continue the SESAM developments with the Derivative approach.

The Optimal and Derivative approaches yield predictions at low microbial biomass that differ from the predictions of the Relative approach. Specifically, they predict that enzymes with low revenue are not expressed at low microbial biomass. This finding generated the constrained enzyme hypothesis for the preservation of organic matter in soils.

*Code availability.* SESAM (v3.1) is available coded in R at https://github.com/bgctw/sesam (last access: February 19th 2024) (doi: 10.5281/zenodo.8026318) and coded in Julia at https://github.com/bgctw/Sesam.jl (last access: February 19th 2024) (doi:10.5281/zenodo.8026366). R source code is released using the GPL-2 licence, because it uses other GPL libraries. Julia code is released using the more permissive MIT License.

The simulation experiments are part of the R repository. They use the derivSesam3P model variant. The Prescribed potential returns code is provided in "Allocation" section of file develop/23_optimAlloc/sesamess/sesam_LRP_deriv.Rmd. The Decadal-term FACE code is provided with file develop/23_optimAlloc/Face1_3P.Rmd. The Seaonsal Incubation code is provided with file SimBareSoilPulse_opt.Rmd.

## Appendix A:  Additional figures

This section provides figures that detail some of the results and provide consistent presentation of main quantities across the experiments. The consistent presentation of all the quantities can not avoid some overplotting.

First, predictions of the FACE simulations experiment for non-normalized residue pool, $R$, and for additional scenarios with adjusted decomposition parameters are shown in Fig. A1.

Next, figures A2, A3, and A4 present common quantities across experiments. They also include a "Fixed" scenario, where enzyme allocation is not adaptive but constant, where specifying the initial value corresponds to specifying another model parameter.

## Appendix B:  Optimal enzyme allocation

This section derives explicit formulas of optimal enzyme allocation by finding the allocation that maximizes total return. It starts with a lemma that states conditions for which the optimum is attained when derivatives are equal. The lemma is then used in subsequent derivations of optimal allocation.

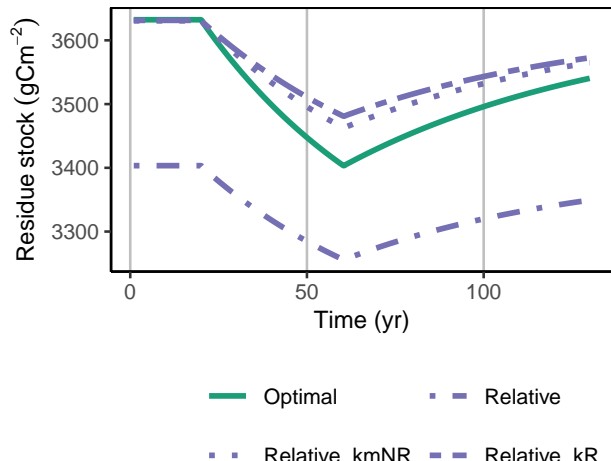

**Figure A1.** Relative approach simulated with a decreased decomposition rate of the residue pool, $k_R$ or an increased $k_{mNR}$, in the FACE simulation experiment, matched the initial steady state stocks but still underestimated the decrease of residue stocks, $R$ $(\mathrm{gCm}^{-2})$, during the period of higher carbon inputs.

## B1 Optima at equality of derivatives

Lemma 1: Let $u_T(\boldsymbol{\alpha}) = C_2 \sum_Z \alpha_Z \operatorname{rev}_Z$ be a function that is a weighted sum of components $\operatorname{rev}_Z$ up to some constant $C_2 \neq 0$, where weights $\alpha_Z \in (0,1)$ add up to one: $\sum_Z^n \alpha_Z = 1$ and component $\operatorname{rev}_Z$ may depend on weight $\alpha_Z$ but not on the other weights. Further, let $\operatorname{rev}_Z$ be differentiable to $\alpha_Z$ and let potential optima $\in (0,1)$. Then at the optima of $u_T(\boldsymbol{\alpha})$ all derivatives $\frac{d(\alpha_Z \operatorname{rev}_Z)}{d\alpha_Z}$ are equal.

Proof: Because of the sum-to-one constraint, we express one of the weights as a function of the other weights and have only $n-1$ free weights.

$$\alpha_n = 1 - \sum_{Z=1}^{n-1} \alpha_Z$$

$$\frac{d\alpha_n}{d\alpha_Z} = -1$$

We are interested in the optima of $u_T$ away from the borders, i.e. $\alpha_{\mathrm{Opt}} \in (0,1)$. In the derivative to $\alpha_Z$ all terms vanish except the term involving $\operatorname{rev}_Z$ and the term involving $\operatorname{rev}_n$, because there $\alpha_n$ is a function of $\alpha_Z$. By the chain rule we have:

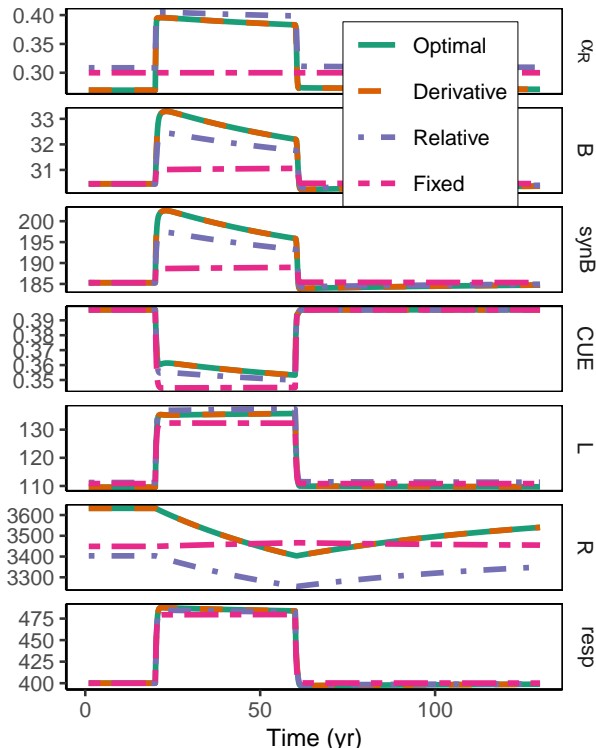

**Figure A2.** Additional quantities and Fixed scenario of the decadal-term experiment compared to Fig. 5. $\alpha_R(\mathrm{gg}^{-1})$: proportion of enzyme allocation to Residue degrading enzyme, $B(\mathrm{gm}^{-2})$: microbial biomass, $\mathrm{syn}_B(\mathrm{gm}^{-2}\mathrm{yr}^{-1})$: C flux for microbial biomass synthesis, $\mathrm{CUE}(\mathrm{gg}^{-1})$: Carbon use efficiency, $L$ and $R$ $(\mathrm{gm}^{-2})$: C in Labile and Residue substrate pool, $\mathrm{resp}(\mathrm{gm}^{-2}\mathrm{yr}^{-1})$: respired C flux. The results based on the Optimal and Drivative are so close together that they overplot.

$$\frac{du_T}{d\alpha_Z} = 0 = C_2\left(\frac{d(\alpha_Z\,\mathrm{rev}_Z)}{d\alpha_Z} + \frac{d(\alpha_n\,\mathrm{rev}_n)}{d\alpha_n}\frac{d\alpha_n}{d\alpha_Z}\right)$$

$$= C_2\left(\frac{d(\alpha_Z\,\mathrm{rev}_Z)}{d\alpha_Z} - \frac{d(\alpha_n\,\mathrm{rev}_n)}{d\alpha_n}\right)$$

Hence, for $C_2 \neq 0$ each $\frac{d(\alpha_Z\,\mathrm{rev}_Z)}{d\alpha_Z}$ has to be equal to $\frac{d(\alpha_n\,\mathrm{rev}_n)}{d\alpha_n}$, i.e. all these derivatives have to be equal.

## B2   Return, revenue, and derivative for a biomineralizing enyzme

The return, revenue and its derivative of a biomineralizing enzyme are slightly more complex than the corresponding quantities of a depolymerizing enzymes (sections 2.1.2 and 2.2.1). They are presented in Table B1 because of the one-columm constraint of normal text in this journal.

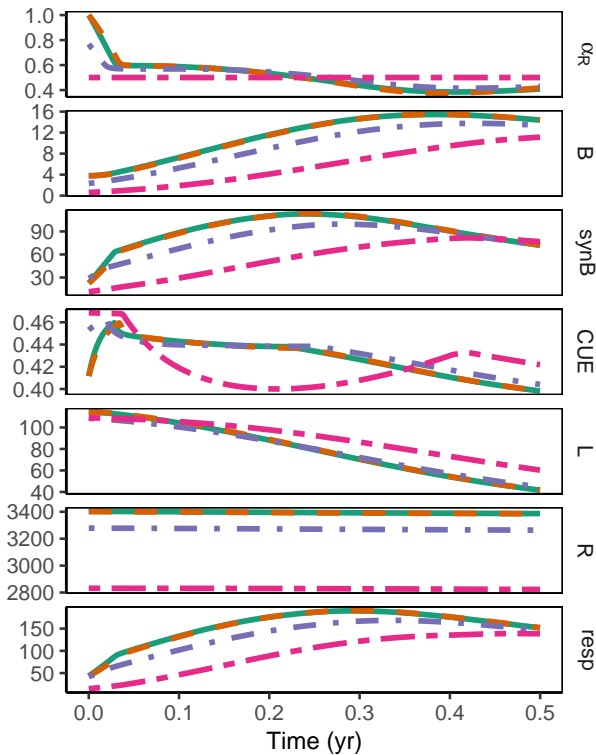

**Figure A3.** Additional quantities and Fixed scenario of the sub-annual experiment compared to Fig. 7. Facets, colors, and linestyles correspond to Fig. A2.

## B3   Explicit optimum formulas

We seek the community composition, here represented by enzyme allocation, $\boldsymbol{\alpha}$, that maximizes total return. This maximizer is located either at the borders of the domain or at a location where all derivatives of the total return are zero. We only look at cases where we know which enzymes take part in the mix with positive allocation, i.e. having $\alpha_Z \in (0, 1)$ and therefore do not need to look at the borders.

The strategy is first to find the small set of allocations where all the derivatives are zero, which includes maxima, minima, and saddle points. Second, we constrain the set to conditions $\alpha_Z \in (0, 1)$ and select that element that results in highest return.

In order to simplify formulas, we make the assumption that all half-saturation parameters are equal: $k_{mNZ} = k_{mN}$.

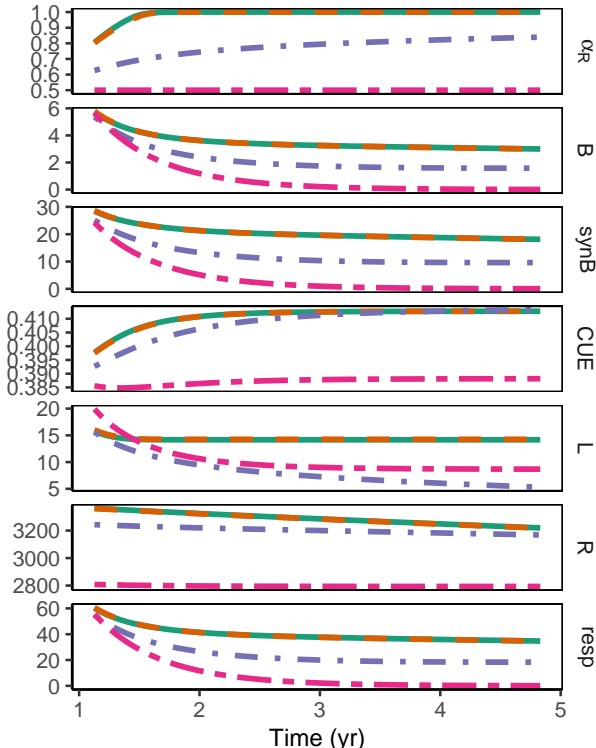

**Figure A4.** Additional quantities and Fixed scenario of the sub-annual experiment compared to Fig. 8. Facets, colors, and linestyles correspond to Fig. A2.

### B3.1 Two depolymerizing enzymes

Utilizing Lemma 1 (Appendix B1) we have:

$$\frac{d(\alpha_L \operatorname{rev}_L)}{d\alpha_L} = \frac{d(\alpha_R \operatorname{rev}_R)}{d\alpha_R}$$

$$\frac{d_L}{(k_{mN} + \alpha_L a_E B)^2} = \frac{d_R}{(k_{mN} + (1-\alpha_L)a_E B)^2}$$

where $\alpha_R = 1 - \alpha_L$

This provides a quadratic equation of $\alpha_L$, which one can solve. We used the Sympy symbolic math tool. That one of the two roots where $\alpha_L \in (0,1)$ and that yields a higher $u_T(\alpha_L)$ provides the optimal $\alpha_L$.

$$\alpha_{L1,2} = \frac{a_E B d_L + k_{mN}(d_L + d_R) \pm \sqrt{d_L d_R}(a_E B + 2k_{mN})}{a_E B (d_L - d_R)}$$

**Table B1.** Equations of return, $d_{Pm}$, revenue, $\mathrm{rev}_P$, and derivative, $\frac{d(\alpha_P\,\mathrm{rev}_P)}{d\alpha_P}$, of a biomineralizing enzyme. Note, that the last one has the same form as the one of the depolymerizing enzyme ($\frac{d(\alpha_Z\,\mathrm{rev}_Z)}{d\alpha_Z}$ in section 2.1.1). It differs, however, in the half-saturation constant of the Michaelis-Menten term which now includes the plant enzyme production: $(e_P + k_{mNP})$.

$$
\begin{aligned}
d_{Pm} &= d_P \frac{e_P + \alpha_P a_E B}{k_{mNP} + e_P + \alpha_P a_E B} - d_P \frac{e_P}{k_{mNP} + e_P} \\
&= d_P \frac{(e_P + \alpha_P a_E B)(k_{mNP} + e_P) - e_P(k_{mNP} + e_P + \alpha_P a_E B)}{(k_{mNP} + e_P + \alpha_P a_E B)(k_{mNP} + e_P)} \\
&= d_P \frac{e_P k_{mNP} + \alpha_P a_E B k_{mNP} + e_P^2 + \alpha_P a_E B e_P - (e_P k_{mNP} + e_P^2 + \alpha_P a_E B e_P)}{(k_{mNP} + e_P)^2 + \alpha_P a_E B(k_{mNP} + e_P)} \\
&= d_P \frac{\alpha_P a_E B k_{mNP}}{(k_{mNP} + e_P)^2 + \alpha_P a_E B(k_{mNP} + e_P)} \\
&= d_P \frac{k_{mNP}}{e_P + k_{mNP}} \frac{\alpha_P a_E B}{(e_P + k_{mNP}) + \alpha_P a_E B} \\
\mathrm{rev}_P &= \frac{d_P}{\omega_{Enz}} \frac{k_{mNP}}{e_P + k_{mNP}} \frac{1}{(e_P + k_{mNP}) + \alpha_P a_E B} \\
\frac{d(\alpha_P\,\mathrm{rev}_P)}{d\alpha_P} &= \frac{d_P}{\omega_{Enz}} \frac{k_{mNP}}{e_P + k_{mNP}} \frac{d}{d\alpha_P}\left( \frac{\alpha_P}{(e_P + k_{mNP}) + \alpha_P a_E B} \right) \\
&= \frac{d_P}{\omega_{Enz}} \frac{k_{mNP}}{e_P + k_{mNP}} \frac{(e_P + k_{mNP}) + \alpha_P a_E B - \alpha_P a_E B}{((e_P + k_{mNP}) + \alpha_P a_E B)^2} \\
&= \frac{d_P}{\omega_{Enz}} \frac{k_{mNP}}{(e_P + k_{mNP} + \alpha_P a_E B)^2}
\end{aligned}
$$

### B3.2 Depolymerizing and biomineralizing enzyme

$$
\frac{d(\alpha_L\,\mathrm{rev}_L)}{d\alpha_L} = \frac{d(\alpha_P\,\mathrm{rev}_P)}{d\alpha_P}
$$

$$
\frac{d_L}{(k_{mN} + \alpha_L a_E B)^2} = \frac{d_P}{((e_P + k_{mN}) + (1 - \alpha_L) a_E B)^2}
$$

$$
\alpha_{L1,2} = \frac{(a_E B + e_P + k_{mN}) d_L + k_{mN} d_P \pm \sqrt{d_L d_P}\,(a_E B + e_P + 2 k_{mN})}{a_E B\,(d_L - d_P)}
$$

### B3.3 Two depolymerizing and one biomineralizing enzyme

We set $\alpha_R = 1 - \alpha_L - \alpha_P$ and have equations of table. B2.

That one of the four roots where $\alpha_P \in (0,1)$ and derived $\alpha_L(\alpha_P) \in (0,1)$ and that yields a highest $u_T(\boldsymbol{\alpha})$ provides the optimal $\boldsymbol{\alpha}$.

### B4   Excursion: replacing revenue by relative profit

Revenue, here, is defined as return per investment, $\mathrm{rev}_Z = d_{Zw}/\mathrm{inv}_{Zw}$. One could argue that one should rather maximize the
profit, i.e. return - investment and corresponding profit revenue, $\mathrm{rev}_{pZ}$, i.e. profit/investment by optimizing enzyme allocation. Here we show, that optimizing the profit yields the same optimal allocation as optimizing the return.

**Table B2.** Potential optima for two depolymerizing and one biomineralizing enzyme

$$\frac{d(\alpha_L \operatorname{rev}_L)}{d\alpha_L} = \frac{d(\alpha_R \operatorname{rev}_R)}{d\alpha_R} = \frac{d(\alpha_P \operatorname{rev}_P)}{d\alpha_P}$$

$$\frac{d_L}{(k_{mN} + \alpha_L a_E B)^2} = \frac{d_R}{(k_{mN} + (1 - \alpha_L - \alpha_P) a_E B)^2} = \frac{d_P}{((e_P + k_{mN}) + \alpha_P a_E B)^2}$$

We first compute $\alpha_L$ given $\alpha_P$ using the first equality.

$$\alpha_{L1,2} = \frac{a_E B (1 - \alpha_P) d_L + k_{mN}(d_L + d_R) \pm \sqrt{d_L d_R}(a_E B(1 - \alpha_P) + 2 k_{mN})}{a_E B(d_L - d_R)}$$

Next we insert the both roots of $\alpha_L(\alpha_P)$ in equating the first and third utility to solve for $\alpha_P$.

For the first root of $\alpha_L$ we get:

$$\alpha_{P1,2} = (A_1 \pm D_1)/B_1$$

$$\begin{aligned}
A_1 = {}& 2Bad_L^{\frac{3}{2}} d_P \sqrt{d_R} - Bad_L^2 d_P - Bad_L d_P d_R + 4d_L^{\frac{3}{2}} d_P \sqrt{d_R} k_{mN} - d_L^3 e_P - d_L^3 k_{mN} \\
& - 2d_L^2 d_P k_{mN} + 2d_L^2 d_R e_P + 2d_L^2 d_R k_{mN} - 2d_L d_P d_R k_{mN} - d_L d_R^2 e_P - d_L d_R^2 k_{mN}
\end{aligned}$$

$$D_1 = \sqrt{d_P}(Ba + e_P + 3k_{mN})\sqrt{-2d_L^{\frac{9}{2}}\sqrt{d_R} + 4d_L^{\frac{7}{2}} d_R^{\frac{3}{2}} - 2d_L^{\frac{5}{2}} d_R^{\frac{5}{2}} + d_L^5 - d_L^4 d_R - d_L^3 d_R^2 + d_L^2 d_R^3}$$

$$B_1 = Ba\left(2d_L^{\frac{3}{2}} d_P \sqrt{d_R} + d_L^3 - d_L^2 d_P - 2d_L^2 d_R - d_L d_P d_R + d_L d_R^2\right)$$

For the second root of $\alpha_L$ we get:

$$\alpha_{P3,4} = (A_2 \pm D_2)/B_2$$

$$\begin{aligned}
A_2 = {}& 2Bad_L^{\frac{3}{2}} d_P \sqrt{d_R} + Bad_L^2 d_P + Bad_L d_P d_R + 4d_L^{\frac{3}{2}} d_P \sqrt{d_R} k_{mN} + d_L^3 e_P + d_L^3 k_{mN} \\
& + 2d_L^2 d_P k_{mN} - 2d_L^2 d_R e_P - 2d_L^2 d_R k_{mN} + 2d_L d_P d_R k_{mN} + d_L d_R^2 e_P + d_L d_R^2 k_{mN}
\end{aligned}$$

$$D_2 = \sqrt{d_P}(Ba + e_P + 3k_{mN})\sqrt{2d_L^{\frac{9}{2}}\sqrt{d_R} - 4d_L^{\frac{7}{2}} d_R^{\frac{3}{2}} + 2d_L^{\frac{5}{2}} d_R^{\frac{5}{2}} + d_L^5 - d_L^4 d_R - d_L^3 d_R^2 + d_L^2 d_R^3}$$

$$B_2 = Ba\left(2d_L^{\frac{3}{2}} d_P \sqrt{d_R} - d_L^3 + d_L^2 d_P + 2d_L^2 d_R + d_L d_P d_R - d_L d_R^2\right)$$

$$\operatorname{rev}_{pZ} = (d_{Zw} - \operatorname{inv}_{Zw})/\operatorname{inv}_{Zw} = \operatorname{rev}_Z - 1$$

$$\frac{d(\alpha_Z \operatorname{rev}_{pZ})}{d\alpha_Z} = \frac{d(\alpha_Z \operatorname{rev}_Z)}{d\alpha_Z} - \frac{d\alpha_Z}{d\alpha_Z} = \frac{d(\alpha_Z \operatorname{rev}_Z)}{d\alpha_Z} - 1$$

The total profit is the sum of profit revenues multiplied by total enzyme investment, $\operatorname{inv}_w$.

$$u_{Tp}(\boldsymbol{\alpha}) = \operatorname{inv}_w \sum_Z \alpha_Z \operatorname{rev}_{pZ}(\alpha_Z)$$

This equation fulfills the conditions of Lemma 1 (Appendix B1) and at the optima all derivatives are equal.

$$\frac{d(\alpha_i \text{rev}_{p_i})}{d\alpha_i} = \frac{d(\alpha_j \text{rev}_{p_j})}{d\alpha_j}$$

$$\frac{d(\alpha_i \text{rev}_i)}{d\alpha_i} - 1 = \frac{d(\alpha_j \text{rev}_j)}{d\alpha_j} - 1$$

$$\frac{d(\alpha_i \text{rev}_i)}{d\alpha_i} = \frac{d(\alpha_j \text{rev}_j)}{d\alpha_j}$$

The last line corresponds to the same condition as when optimizing returns. Hence, they lead to the same optima.

## Appendix C:  Derivation of the relative approach

The Relative approach approximates optimal allocation by setting optimal allocation proportional to revenue (2.2.3). Hence, we seek the conditions for which the following relationship holds:

$$\frac{\alpha_j}{\alpha_i} \approx \frac{\text{rev}_j}{\text{rev}_i}$$

At the solution of the Optimal approach all the derivatives of (revenue times $\alpha$) for all enzymes in the mix are equal (Appendix 2.2.1). By using $\frac{d(\alpha_Z \text{rev} Z)}{d\alpha_Z} \approx \text{rev}_Z \frac{e_Z + k_{mNZ}}{\alpha_Z a_E B}$, as shown below, for any two enzymes $i, j$ we have:

$$\frac{d(\alpha_i \text{rev}_i)}{d\alpha_i} = \frac{d(\alpha_j \text{rev}_j)}{d\alpha_j}$$

$$\text{rev}_i \frac{e_i + k_{mNi}}{\alpha_i a_E B} \approx \text{rev}_j \frac{e_j + k_{mNj}}{\alpha_j a_E B}$$

$$\frac{\alpha_j}{\alpha_i} \approx \frac{\text{rev}_j}{\text{rev}_i} \frac{e_j + k_{mNj}}{e_i + k_{mNi}}$$

$$\frac{\alpha_j}{\alpha_i} \approx \frac{\text{rev}_j}{\text{rev}_i}$$

The last approximation holds only for similar half-saturation parameters across enzymes $k_{mNZ} \approx k_{mN}$, and plant enzyme production being low compared to this half-saturation: $e_Z \ll k_{mN}$.

The first approximation in the second line is only valid for an enzyme production flux that is not larger than the half-saturation, $k_{mNZ}$ (see below). This is violated at low microbial biomass or very low $\alpha_Z$.

Hence, the optimal allocation is approximately proportional to the revenue for the combination of the following conditions:

– all enzymes have a non-negligible share

– microbial biomass is sufficiently high

– plant biomineralizing enzyme production is low.

The derivation above used the following relationship that still needs to be shown: $\frac{d(\alpha_Z \operatorname{rev}_Z)}{d\alpha_Z} \approx \operatorname{rev}_Z \frac{e_Z + k_{mNZ}}{\alpha_Z a_E B}$.

For depolymerizing enzymes we use the following approximations. For $\alpha_Z a_E B \gg k_{mNZ}$, i.e. $2k_{mNZ} + \alpha_Z a_E B \approx \alpha_Z a_E B$,

the half-saturation $k_{mNZ}$ can be neglected in the denominator of the revenue. Note that $\alpha_Z a_E B \gg k_{mNZ}$ implies $(\alpha_Z a_E B)^2 \gg k_{mNZ}^2$.

$$
\begin{aligned}
\operatorname{rev}_Z &= \frac{d_Z}{\omega_{Enz}} \frac{1}{k_{mNZ} + \alpha_Z a_E B} \\
&\approx \frac{d_Z}{\omega_{Enz}} \frac{1}{\alpha_Z a_E B} \\
\frac{d(\alpha_Z \operatorname{rev}_Z)}{d\alpha_Z} &= \frac{d_Z}{\omega_{Enz}} \frac{k_{mNZ}}{(k_{mNZ} + \alpha_Z a_E B)^2} \\
&= \frac{d_Z}{\omega_{Enz}} \frac{k_{mNZ}}{k_{mNZ}^2 + 2k_{mNZ}\alpha_Z a_E B + (\alpha_Z a_E B)^2} \\
&\approx \frac{d_Z}{\omega_{Enz}} \frac{k_{mNZ}}{\alpha_Z a_E B (2k_{mNZ} + \alpha_Z a_E B)} \\
&= \operatorname{rev}_Z \frac{k_{mNZ}}{2k_{mNZ} + \alpha_Z a_E B} \\
&\approx \operatorname{rev}_Z \frac{k_{mNZ}}{\alpha_Z a_E B}
\end{aligned}
$$

where the first two relationships have been derived in Appendix 2.1.1. For depolymerizing enzymes we have, $e_Z = 0$, because they are not produced by plant roots.

Similarly, for biomineralizing enzymes we require $\alpha_Z a_E B \gg k_{mNZ} + e_Z$, where $e_Z$ is the production of enzyme $Z$ by plants.

$$
\begin{aligned}
\operatorname{rev}_Z &= \frac{d_Z}{\omega_{Enz}} \frac{k_{mNZ}}{e_Z + k_{mNZ}} \frac{1}{e_Z + k_{mNZ} + \alpha_Z a_E B} \\
&\approx \frac{d_Z}{\omega_{Enz}} \frac{k_{mNZ}}{e_Z + k_{mNZ}} \frac{1}{\alpha_Z a_E B} \\
\frac{d(\alpha_Z \operatorname{rev}_Z)}{d\alpha_Z} &= \frac{d_Z}{\omega_{Enz}} \frac{k_{mNZ}}{(e_Z + k_{mNZ} + \alpha_Z a_E B)^2} \\
&= \frac{d_Z}{\omega_{Enz}} \frac{k_{mNZ}}{(e_Z + k_{mNZ})^2 + 2(e_Z + k_{mNZ})\alpha_Z a_E B + (\alpha_Z a_E B)^2} \\
&\approx \frac{d_Z}{\omega_{Enz}} \frac{k_{mNZ}}{\alpha_Z a_E B (2(e_Z + k_{mNZ}) + \alpha_Z a_E B)} \\
&= \operatorname{rev}_Z \frac{e_Z + k_{mNZ}}{2(e_Z + k_{mNZ}) + \alpha_Z a_E B} \\
&\approx \operatorname{rev}_Z \frac{e_Z + k_{mNZ}}{\alpha_Z a_E B}
\end{aligned}
$$

## Appendix D: Derivative-based change of community allocation

SESAM assumes that microbial community develops in a way to maximize growth of the entire community. Growth increases
with uptake and hence increases with decomposition flux for given enzyme allocation. The revenue of allocation to enzyme

$Z$ is $\mathrm{rev}_Z = \frac{d_{Zw}(\alpha_Z)}{\alpha_Z \omega_{Enz} a_E B}$. The return $d_{Zw}$ is a limitation-weighted mineralization flux or uptake flux of nutrients and carbon (sections 2.1.1 and 2.1.2). The investment is the share, $\alpha_Z$, invested into production of enzyme $Z$ multiplied by total limitation-weighted flux, $\omega_{Enz} a_E B$, allocated to enzyme production.

Although it is possible to derive explicit formula for the allocation that optimizes total return for simple cases, the formulas quickly grow and involve higher-order polynomials of $\alpha$ with several solutions outside the reasonable bound $\alpha_Z \in [0,1]$.

Here we follow an alternative local approach were we assume the rate change of $\alpha_Z$ over time to be proportional to the deviation of the derivative of change of total return with respect to $\alpha_Z$ to the average across the derivatives for different enzymes. The higher the increase in total return for shifting allocation towards a specific enzyme, the faster the community changes in this direction.

The total return is a weighted sum of enzyme revenues, and derivatives of $\frac{d(\alpha_Z \mathrm{rev}_Z)}{d\alpha_Z}$ have been derived for depolymerizing and biomineralizing enzymes (section 2.2.1).

$$u_T = \omega_{Enz} a_E B \sum_Z \alpha_Z \mathrm{rev}_Z(\alpha_Z)$$
$$\frac{du_T}{d\alpha_Z} = \omega_{Enz} a_E B \sum_Z \frac{d(\alpha_Z \mathrm{rev}_Z)}{d\alpha_Z}$$

We assume that the larger the change in return with increasing allocation, i.e. the derivative to allocation coefficient $\alpha_Z$, the larger is the change in allocation. In addition to the assumption of proportionality to the derivative, we assume that the community changes at a rate of the same magnitude as synthesis and turnover of microbial biomass.

$$\frac{d\alpha_Z}{dt} \propto \frac{du_T}{d\alpha_Z} - m_{du}$$
$$= \left( \frac{|\mathrm{syn}_B|}{B} + \tau \right) \frac{\frac{du_T}{d\alpha_Z} - m_{du}}{m_{du}}$$
$$m_{du} = \mathrm{mean}_i \left( \frac{du_T}{d\alpha_i} \right)$$

where $m_{du}$ is the average across derivatives of return across enzymes that are allocated to. If all changes are the same, i.e. equal to the mean, the allocation is optimal because it does not increase in any direction.

We want the change to be proportional to the change in return compared to the average return. Subtracting this mean ensures that the sum of all the changes in $\alpha$ sums to zero so that the sum across $\alpha$ is preserved. The proportionality factor normalizes the change in return and multiplies this relative change by the rate of microbial turnover, composed of biomass synthesis and biomass turnover.

## D1 Exclude enzymes whose negative relative change is larger than its share

Community may not allocate to all enzymes. Hence, $m_{due}$ (an updated version of $m_{du}$) averages only across a subset of enzymes. The derivative optimization strategy assumes that nothing is allocated to an enzyme if its normalized change towards zero is larger than than its current share, i.e. is more negative than $-\alpha_Z$.

$$Z_0 = \left\{ Z \Big| \frac{\frac{du_T}{d\alpha_Z} - m_{due}}{m_{due}} < -\alpha_Z \right\}$$

$$\frac{d\alpha_Z}{dt} = \left( \frac{|\mathrm{syn}_B|}{B} + \tau \right) \begin{cases} -\alpha_Z & \text{for } Z \in Z_0 \\ \frac{\frac{du_T}{d\alpha_Z} - m_{due}}{m_{due}} & \text{otherwise} \end{cases}$$

$$= \left( \frac{|\mathrm{syn}_B|}{B} + \tau \right) \max \left( \frac{\frac{du_T}{d\alpha_Z} - m_{due}}{m_{due}}, -\alpha_Z \right)$$

$$m_{due} = \frac{\sum_{\zeta \notin Z_0} \frac{du_T}{d\alpha_\zeta}}{|\{Z\} \setminus Z_0| + \sum_{\zeta \in Z_0} \alpha_\zeta}$$

Where $|\{Z\} \setminus Z_0|$ denotes the number of enzymes allocated to, i.e. the cardinality of the set of all enzymes without those in $Z_0$ The relative change of those excluded enzymes is set to $-\alpha_Z$, resulting in negative changes going to zero as $\alpha_Z$ approaches zero. Hence, the relative change is lower-bounded by $-\alpha_Z$.

$m_{du}$ has to be adjusted to $m_{due}$, so that $\sum_i \frac{d\alpha_i}{dt} = 0$ holds.

$$\sum_{\zeta \notin Z_0} \frac{\frac{du_T}{d\alpha_Z} - m_{due}}{m_{due}} + \sum_{\zeta \in Z_0} -\alpha_\zeta = 0$$

This definition is recursive, because $m_{due}$ is computed across a set that is defined using $m_{due}$. In order to determine $Z_0$ one can start with the empty set and add all enzymes that fulfill the condition. If enzymes were added then the mean across remaining derivatives increases, and the condition has to be checked again. Hence, adding enzymes to $Z_0$ is repeated until no more enzymes fulfill the condition and the mean does not change any more.

*Author contributions.* TW and CR developed the math, TW implemented it into the model, and led the writing of the manuscript. TW, CR, BA, and MS contributed to the discussion of results and writing of the manuscript

*Competing interests.* The contact author has declared that none of the authors has any competing interests.

*Acknowledgements.* We thank Lin Yu for fruitful discussion. We thank the Max Planck Society for funding.

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
