# Peer review of "Optimal enzyme allocation leads to the constrained enzyme hypothesis: The Soil Enzyme Steady Allocation Model (SESAM v3.1)."

_EGUsphere, 2023_

## Author Comment (AC1)

We thank Stefano Manzoni for his encouraging feedback. In this answer we repeat the comments with a label (Reviewer comment RC1) for reference before each of our replies (author comment AC) in blue.

RC1-1: The manuscript by Wutzler and co-authors presents a theoretical analysis of three approaches to define extracellular enzyme allocation by soil microbes. The topic is timely and important given the interest in developing microbial explicit models to predict carbon and nutrient cycling in soils. These three approaches build on concepts presented in earlier publications by Wutzler and co-authors, but are of sufficient novelty to warrant publication in a separate article. Specifically, the three approaches are based on the hypothesis that enzymes are optimally allocated to ensure maximum (instantaneous) microbial growth at the whole community level, but differ in the way optimization is implemented— from a rigorous maximization of total return to approximated relations that are easier to implement in models.

I do not have major concerns regarding the model setup and concept, but rather some comments to improve the presentation and clarify the model rationale and derivations.

Main comments

RC1-2 Readers not familiar with the SESAM model will find it difficult to understand how the proposed developments fit into the broader model. I would add a model schematic in Section 2.1, pointing to the components in the model that affect (and are affected by) enzyme allocation. This could be a place to also define (graphically) the main fluxes used in the mathematical derivations.

AC1-2: Thank you for this suggestions. We will extend the model description, add a schematic and highlight the changes to the previous version. In addition, we will extend the model description and move symbols and some of the appendices to the main text.

RC1-3: Maximization criterion: I agree that community-level maximization makes sense in practice, but I wonder about theoretical support for that (this is something I am also struggling with!). In a microbial community where interactions are dominated by facilitation, maximization of fitness of individual taxa might lead to maximization of community level growth, but what would happen in highly competitive environments? In section 4.3 there could be space for a short comment on the applicability of community-level maximization criteria.

AC1-3: Thank you for these suggestions. This is an entire new issue that we think warrants its own discussion paper and therefore wrote in the introduction "The arguments why this optimality assumption is valid are rooted in systems ecology, which focuses on the entire system rather than specific interactions. But this discussion goes beyond to scope of this manuscript and here we assume that it is promising to explore the assumption that growth of the entire community is optimized.". Nevertheless, we will make an attempt to come up with a concise introduction and/or discussion of this topic.

RC1-4: Presentation of main quantities: it would be easier to understand the theory if the main quantities were presented in mathematical form in the main text, whereas now they are spread between text and appendices. I would suggest providing equations and explanations for revenue, return, enzyme

investment, microbial growth and other relevant fluxes in Section 2.2, using material now in the appendices. I would also suggest defining the element limitation weights, that are often mentioned but not explained nor listed in the symbol tables.

AC1-4: In the revised manuscript we will move appendices A and C2 to the main text. We hope to still strike the difficult balance between presenting the overall flow of the argument and the details. We will restate and explain the formula of limitation weights and put several more quantities in the "Further symbols" table.

RC1-5: Control simulations without adaptation: good idea to include a no-adaptation scenario in some analyses, but why not showing it in all figures, to give an idea of the effect of dynamic allocation compared to a 'control' scenario?

AC1-5: We will do this. However, we will only present it in the Appendix or supplement in probably several figures. The reason is that differences between optimization approaches, that we are interested in here, are much smaller than differences to the non-adapting version. Hence, they are better visualized when not showing the non-adaptive version at the same axis.

RC1-6: Shown variables: it would be helpful to show the same quantities in all (or most) figures, so readers can better appreciate how different numerical experiments affect the dynamics of the same quantities. For example, total respiration and microbial growth, as well as carbon in key model compartments could be shown almost in all figures (they are already shown in some). CUE is now only shown in a figure in appendix, but it is discussed in depth in the main text, so I would suggest showing it in the main text as well.

AC1-6: We will move the CUE figure from appendix to the main text. We will provide figures of common quantities across experiments in a supplement. So far in each figure, we focused on a specific message, which is harder to convey if showing more quantities. Additionally, the figure captions are already at the too-long side. Most of the quantities that change with the decadal-term experiment are not changing very much with the shorter-term experiments. Showing those figures in the main text probably will be distracting.

RC1-7: Section 4.2: this is a very interesting discussion point and a key overall message. The flip side of the constrained enzyme hypothesis is that also low returns due to very diluted substrates would result in low production of enzymes and thus accumulation of those substrates (https://doi.org/10.1038/s41561-020-0612-3). Similar ideas had been proposed earlier in marine biogeochemistry and referred to as 'dilution hypothesis' (https://www.science.org/doi/10.1126/science.1258955). Perhaps this section could be supported a bit more using literature on aquatic systems, where return on enzyme investment has been studied for some time.

AC1-7: Thanks for hinting us to the aquatic literature and other soil literature. We already discussed the first reference and the dilution/selective preservation hypothesis in the manuscript: L242ff "It differs from the selective preservation hypothesis (Lehmann and Kleber, 2015) by making preservation dependent on the size or density of microbial biomass and by making preservation dependent on the availability of alternative substrates. When microbial biomass grows, e.g. by adding enough labile substrate, the focus on solely the enzymes with highest revenue is not beneficial any more and the optimal microbial community also invests into decomposition of the organic matter with lower revenue." We will extend this discussion in the revised version much more.

Minor comments

AC: Thanks for the minor comments. We will adapt them. The only exception we answer here:

RC1-8: L62 and elsewhere: I would use the term 'mortality' instead of 'turnover' if the process being modelled is indeed just mortality

AC1-8: 'microbial turnover' flux in SESAM actually includes all losses from the microbial pool in addition to respiration and synthesis of extracellular enzymes. It includes e.g. production of EPS. Hence, 'mortality' is not a good choice. We agree that also 'turnover' is not a very good choice, because it suggests to include all pool turnover fluxes including respiration and enzyme synthesis. Since, we called it 'turnover' in the previous SESAM publications, we rather keep this term for consistency, here.

We will adapted the following comments:

L80-82: I would reverse the order of these sentences, first saying what 'revenue' is and then explaining why it is an important quantity

Figure 1 is difficult to understand. How do we read that \alpha_L increases by looking at the shown curves? The sentence starting "Starting at levels…" is also quite convoluted. Maybe it would help including additional panels showing trends in return and microbial growth rate as a function of \alpha_L?

L89-90: convoluted sentence, hard to understand

L99: point to corresponding equations in the appendices

Figure 2: check units on x-axis (should be "…yr^{-1}"?) and "… labile C…" (missing "C")

Section 2.3.1: please add units next to numerical values

Section 2.3.3: worth adding a short motivation for modelling seasonal processes with a model designed for decadal scale dynamics

L151: suggested edit "In the Incubation experiment, a labile C-depleted soil was amended…"

L162: "Derivative"

Figure 3, caption: units of L should not contain "yr^{-1}"

L168: "By adjusting"

L188-190: convoluted sentence, hard to understand

L209: "should help" to achieve what? Not quite clear

End of P10: I see the point of discussing why certain implementation approaches might be numerically problematic, but it is not easy to follow the arguments without visual support; also, typical time scales for enzyme deactivation and microbial mortality are in the order of weeks to few months, so much shorter than the scale at which SESAM is meant to be run

L284-288: I do not fully agree with this interpretation, as during drought physiological mechanisms lead to slower growth at the community level (some taxa go dormant, others spend energy to maintain turgor via osmolytes…). My impression is that in dry conditions the production of enzymes per se is not the first priority of the microbes

L326: in other words, enzymes do no interact with each other so rev_Z depends only on \alpha_Z—is this a reasonable interpretation?

P15: I like how this derivation is presented—very elegant!

Supplementary tables: please add missing symbols, such as \omega_C, \omega_N, \omega_P, \nu_{TC}, \nu_{TN}, \nu_{TP} (unless the \nu_{TE} are the same as \nu_{T}?)

L346: "this condition implies…"

Figure B2: it would help to show also \alpha values in this figure

P18, equation for d_{ZC} and following equations: is \alpha the same as \alpha_C? Also, the weighing factors \omega are not defined

L393: is \alpha_{P1,2} the same as \alpha_{1,2} in Section C3.1?

Section C4: just a semantic comment, profit is maximized by optimizing enzyme allocation (to be consistent with the rest of the manuscript)

L498, L405: in both lines, "profit/investment", not just "profit" (I think)

L411: "By using…"

L433-434: convoluted sentence, hard to understand

L437: "it is possible…"

We thank Sergey Blagodatsky for his critical and positive feedback. In this answer we repeat the comments with a label (Reviewer comment RC2) for reference before each of our replies (author comment AC) in blue.

RC2-1: The paper by Wutzler et al., "Optimal enzyme allocation leads to the constrained enzyme hypothesis: The Soil Enzyme Steady Allocation Model (SESAM v3.1)" describes the current development of the SESAM model, where three approaches for the calculation of enzyme activity allocation by soil microorganisms were compared. The main conclusion is that the derivative approach proposed in this MS differs from the relative approach and gives more realistic values in model outputs in the case of low microbial biomass levels. The authors argue that this approach is promising and plan to use it in further model development. Based on model predictions (namely from three simulation experiments), the authors put forward the constrained enzyme hypothesis, which they believe explains the long-term persistence of SOM. In the discussion, the authors mention that this hypothesis is complementary to other current explanations of this phenomenon, but I remain sceptical about the relative importance of the constrained enzyme hypotheses in explaining SOM persistence. This view of the authors is not supported by experimental data, and the results of the presented simulation experiments are strongly dependent on the prescribed model parameterisation and the chosen classification of enzyme groups. From section 4.4, which discusses the observational evidence for the proposed hypothesis, it becomes clear that there is no data yet to confirm the constrained enzyme hypothesis.

AC2-1. RC2 correctly pointed out, that the constrained enzyme hypothesis is only a hypothesis that is currently not backed up by experimental data but only an insight depending on the SESAM modelling assumptions (although not on a specific parametrization). The revised manuscript already made this clear in section 4.4: "The constrained enzyme hypothesis is a consequence of several model assumptions. It was derived without reference to observed patterns.". Rather than believing in this hypothesis we intent to provide a possible explanation for some of the observed patterns that differs from the explanations given so far and will extend the corresponding discussion section.

RC2-2 Figure 6 illustrates the spatial separation of substrates and enzymes at low biomass levels in the soil (right part) rather than the production of fewer enzyme types.

AC2-2: We did not encounter this unintended interpretation of Figure 6 before. Note that in the right part one type of enzymes is completely absent. Currently, we have no idea of better visualizing the hypothesis and intent to clarify in an updated figure caption.

RC2-3 However, these remarks do not imply that the work is of low quality or incomplete. The idea of allocating resources to the production of different enzyme groups as a response of the microbial community to specific limitation by one or two key nutrients is very valuable and timely. The current MS provides an excellent basis for further solutions in modelling soil microbial biomass and SOM dynamics. The next step would be to test the new model version against experimental data, where the

current division into enzyme groups could be confirmed or rejected with suggestions for further model structure modifications.

AC2-3: Thank you for this positive feedback. We are indeed working on a publication where we test the model by fitting it to data from sites of a phosphorus gradient.

Specific comments and technical notes
RC2-4: Section 2.1: I would suggest providing a model scheme for readers who are facing the SESAM model for the first time. In order not to exactly repeat the scheme of Wutzler et al. 2022, it would be desirable to show the parts (or application points of allocation optimisation approaches) newly developed in this study. It is also not clear from which pool P is obtained. In the 2022 paper there are two enzyme groups for P and in your current publication there seems to be 1, or?

AC2-4: We will extend the introduction of the SESAM model. Since, this is a cumulative model description series we will repeat only the most relevant parts of the previous model description paper. The version 3.0 described in Wutzler et al. 2022 had no P cycle. However, we derived and compared the optimality approaches already for a P-enabled upcoming version. The two depolymerizing enzymes release P in stoichiometric quantities of both, the L and R source pools. However, there is and additional class of P-biomineralizing enzymes are conceptually different from the depolymerizing enzymes. They can cleave P from the substrates without making C and N available in substrate-stoichiometric quantities at the same time. Therefore, it was important to develop the equations already for this extended case.

RC2-5: L150: There is a misprint, please correct. Seasonal does not seem to be the right term to describe this simulation experiment. Does the temperature and the moisture vary in this case? If not, perhaps 6-month incubation would be a more accurate name for this part.

AC2-5: The simulation experiment did not take into account temperature and moisture effects. This corresponds to holding model drivers other than litter inputs constant. In the earth-system modelers community the focus scale of this experiment that is longer than weekly and shorter than annually, is called "seasonal". Nevertheless, we will change it to "sub-annual".

RC2-6: L151: Please edit the sentence, as it is not fully clear.

AC2-6: We will split the sentence: "The Incubation simulation experiment added a portion of labile litter to a previously labile-depleted soil. Next, it tracked the carbon use efficiency (CUE) of the microbial community over time and across different C:N ratios of the added labile organic matter."

RC2-7: Fig.2. The legend on the figure can be expanded to make it easier to understand. The meaning of R, L and P can be deduced from the method section, where the numerical experiment is described, but a direct description, i.e. residues, labile and phosphorus-targeted enzymes, would improve the figure.

AC2-7: Thanks for noting. We will extend the figure caption.

---

## Author Response (AR1)

We thank Stefano Manzoni for his encouraging feedback. In this answer we repeat the comments with a label (Reviewer comment RC1) for reference before each of our replies (author comment AC) in blue.

RC1-1: The manuscript by Wutzler and co-authors presents a theoretical analysis of three approaches to define extracellular enzyme allocation by soil microbes. The topic is timely and important given the interest in developing microbial explicit models to predict carbon and nutrient cycling in soils. These three approaches build on concepts presented in earlier publications by Wutzler and co-authors, but are of sufficient novelty to warrant publication in a separate article. Specifically, the three approaches are based on the hypothesis that enzymes are optimally allocated to ensure maximum (instantaneous) microbial growth at the whole community level, but differ in the way optimization is implemented—from a rigorous maximization of total return to approximated relations that are easier to implement in models.

I do not have major concerns regarding the model setup and concept, but rather some comments to improve the presentation and clarify the model rationale and derivations.

Main comments

RC1-2 Readers not familiar with the SESAM model will find it difficult to understand how the proposed developments fit into the broader model. I would add a model schematic in Section 2.1, pointing to the components in the model that affect (and are affected by) enzyme allocation. This could be a place to also define (graphically) the main fluxes used in the mathematical derivations.

AC1-2:  We added two schematic figures and in the first figure highlighted the changes to the previous SESAM version. In addition, we extended the method section by restating decomposition and return fluxes, as well as elemental limitation. Moreover, we moved several parts of the appendices including symbol-tables to the main text. (section 2.1)

RC1-3: Maximization criterion: I agree that community-level maximization makes sense in practice, but I wonder about theoretical support for that (this is something I am also struggling with!). In a microbial community where interactions are dominated by facilitation, maximization of fitness of individual taxa might lead to maximization of community level growth, but what would happen in highly competitive environments? In section 4.3 there could be space for a short comment on the applicability of community-level maximization criteria.

AC1-3: We  wrote a concise summary sketching the main line of arguments in the introduction. For brevity it is necessarily quite abstract. (L 38ff)

RC1-4: Presentation of main quantities: it would be easier to understand the theory if the main quantities were presented in mathematical form in the main text, whereas now they are spread between text and appendices. I would suggest providing equations and explanations for revenue, return, enzyme investment, microbial growth and other relevant fluxes in Section 2.2, using material now in the appendices. I would also suggest defining the element limitation weights, that are often mentioned but not explained nor listed in the symbol tables.

AC1-4: In the revised manuscript we moved appendices A and C2 to the main text, which present return, revenue, enzyme investment, and now also limitation weights. We updated the symbols table to denote limitation weights and the elemental-limitation factors (section 2.1). We still kept the equations of computing the optima in own appendices to not overwhelm the main text with equations.

RC1-5: Control simulations without adaptation: good idea to include a no-adaptation scenario in some analyses, but why not showing it in all figures, to give an idea of the effect of dynamic allocation compared to a 'control' scenario?

AC1-5: We included a no-adaptation scenario in the FACE experiment.  We however, did not show it in all figures but rather in additional Appendix figures (Appendix A). The reason is, that differences between optimization approaches, that we are interested in here, are much smaller than differences to the non-adapting version. Hence, they are better visualized when not showing the non-adaptive version at the same axis.
For the Prescribed potential returns a no-adaptation scenario does not make sense, because in this experiment all the states variables, and hence also fluxes, are fixed. Presenting a constant allocation factor over varying potential return does not add insight.

RC1-6: Shown variables: it would be helpful to show the same quantities in all (or most) figures, so readers can better appreciate how different numerical experiments affect the dynamics of the same quantities. For example, total respiration and microbial growth, as well as carbon in key model compartments could be shown almost in all figures (they are already shown in some). CUE is now only shown in a figure in appendix, but it is discussed in depth in the main text, so I would suggest showing it in the main text as well.

AC1-6: We moved the CUE figure from appendix to the main text and added a facet-row of allocation alpha_R to it (Fig 6). We provided consistent figures of common quantities across experiments in Appendix A. Wit showing all the quantities there is necessarily much overplotting. We decided to keep the main text figures to only a subset of the quantities so that they can convey a specific message, which is harder to convey if showing more quantities. Moreover, their figure captions are already quite long.

RC1-7: Section 4.2: this is a very interesting discussion point and a key overall message. The flip side of the constrained enzyme hypothesis is that also low returns due to very diluted substrates would result in low production of enzymes and thus accumulation of those substrates (https://doi.org/10.1038/s41561-020-0612-3). Similar ideas had been proposed earlier in marine biogeochemistry and referred to as 'dilution hypothesis' (https://www.science.org/doi/10.1126/science.1258955). Perhaps this section could be supported a bit more using literature on aquatic systems, where return on enzyme investment has been studied for some time.

AC1-7: Thanks for hinting us to the aquatic literature and other soil literature. We already shortly discussed the first reference and the dilution/selective preservation hypothesis of the Lehmann paper in the previous manuscript. Now we extended this discussion with also referencing the aquatic literature (L 341ff).

Moreover, we added an entire new section discussion our results in comparison to a recently published study (Weverka et a. 2023) which used similar assumptions and yielded similar conclusions (L 348ff).

Minor comments

AC: Thanks for the minor comments. We adopted them. The only two exception we answer here:

RC1-8: L62 and elsewhere: I would use the term 'mortality' instead of 'turnover' if the process being modelled is indeed just mortality

AC1-8: 'microbial turnover' flux in SESAM actually includes all losses from the microbial pool in addition to respiration and synthesis of extracellular enzymes. It includes e.g. production of EPS. Hence, 'mortality' is not a good choice. We agree that also 'turnover' is not a very good choice either, because it suggests to include all pool turnover fluxes including respiration and enzyme synthesis. Since, we called it 'turnover' in the previous SESAM publications, we rather keep this term for consistency, here.

RC1-8: L326: in other words, enzymes do no interact with each other so rev_Z depends only on \alpha_Z—is this a reasonable interpretation?

AC1-9: (Refers to appendix B1: "component revZ may depend on weight $\alpha Z$ but not on the other weights")
Yes. If a specific enzyme action depended also on the levels of the other enzymes this assumptions is violated. Such a dependence could become important if different enzymes compete for the same binding sites of the same substrate in a well mixed soil solution. However, SESAM models enzyme classes for specific substrate classes and and no common target substrates. Moreover, it envisions the degradation of different substrates in a way where enzymes are spatially located near their producers and target substrates, i.e. constraints to mixing, while dissolved products and especially the inorganic nutrients are assumed to diffuse and mix better.

We adapted the following comments:

L80-82: I would reverse the order of these sentences, first saying what 'revenue' is and then explaining why it is an important quantity

Figure 1 is difficult to understand. How do we read that \alpha_L increases by looking at the shown curves? The sentence starting "Starting at levels…" is also quite convoluted. Maybe it would help including additional panels showing trends in return and microbial growth rate as a function of \alpha_L?

L89-90: convoluted sentence, hard to understand

L99: point to corresponding equations in the appendices

Figure 2: check units on x-axis (should be "…yr^{-1}"?) and "… labile C…" (missing "C")

Section 2.3.1: please add units next to numerical values

Section 2.3.3: worth adding a short motivation for modelling seasonal processes with a model designed for decadal scale dynamics

L151: suggested edit "In the Incubation experiment, a labile C-depleted soil was amended…"

L162: "Derivative"

Figure 3, caption: units of L should not contain "yr^{-1}"

L168: "By adjusting"

L188-190: convoluted sentence, hard to understand

L209: "should help" to achieve what? Not quite clear

End of P10: I see the point of discussing why certain implementation approaches might be numerically problematic, but it is not easy to follow the arguments without visual support; also, typical time scales for enzyme deactivation and microbial mortality are in the order of weeks to few months, so much shorter than the scale at which SESAM is meant to be run

L284-288: I do not fully agree with this interpretation, as during drought physiological mechanisms lead to slower growth at the community level (some taxa go dormant, others spend energy to maintain turgor via osmolytes…). My impression is that in dry conditions the production of enzymes per se is not the first priority of the microbes

P15: I like how this derivation is presented—very elegant!

Supplementary tables: please add missing symbols, such as $\omega_C$, $\omega_N$, $\omega_P$, $\nu_{TC}$, $\nu_{TN}$, $\nu_{TP}$ (unless the $\nu_{TE}$ are the same as $\nu_T$?)

L346: "this condition implies…"

Figure B2: it would help to show also $\alpha$ values in this figure

P18, equation for $d_{ZC}$ and following equations: is $\alpha$ the same as $\alpha_C$? Also, the weighing factors $\omega$ are not defined

L393: is $\alpha_{P1,2}$ the same as $\alpha_{1,2}$ in Section C3.1?

Section C4: just a semantic comment, profit is maximized by optimizing enzyme allocation (to be consistent with the rest of the manuscript)

L498, L405: in both lines, "profit/investment", not just "profit" (I think)

L411: "By using…"

L433-434: convoluted sentence, hard to understand

L437: "it is possible…"

We thank Sergey Blagodatsky for his critical and positive feedback. In this answer we repeat the comments with a label (Reviewer comment RC2) for reference before each of our replies (author comment AC) in blue.

RC2-1: The paper by Wutzler et al., "Optimal enzyme allocation leads to the constrained enzyme hypothesis: The Soil Enzyme Steady Allocation Model (SESAM v3.1)" describes the current development of the SESAM model, where three approaches for the calculation of enzyme activity allocation by soil microorganisms were compared. The main conclusion is that the derivative approach proposed in this MS differs from the relative approach and gives more realistic values in model outputs in the case of low microbial biomass levels. The authors argue that this approach is promising and plan to use it in further model development. Based on model predictions (namely from three simulation experiments), the authors put forward the constrained enzyme hypothesis, which they believe explains the long-term persistence of SOM. In the discussion, the authors mention that this hypothesis is complementary to other current explanations of this phenomenon, but I remain sceptical about the relative importance of the constrained enzyme hypotheses in explaining SOM persistence. This view of the authors is not supported by experimental data, and the results of the presented simulation experiments are strongly dependent on the prescribed model parameterisation and the chosen classification of enzyme groups. From section 4.4, which discusses the observational evidence for the proposed hypothesis, it becomes clear that there is no data yet to confirm the constrained enzyme hypothesis.

AC2-1. RC2 correctly pointed out, that the constrained enzyme hypothesis is only a model hypothesis that is currently not backed up by experimental data but only an insight depending on the SESAM modelling assumptions (although not on a specific parametrization).
The revised manuscript makes this exploration of assumptions clear already in new introduction section (L 47): "Hence, the SESAM optimal community growth assumptions is rational, but it is still an assumption to be challenged. "
Additionally it keeps the clarification of section 4.4 (L 386): "The constrained enzyme hypothesis is a consequence of several model assumptions. It was derived without reference to observed patterns." and (L 404): "In summary, studies that specifically look at enzyme diversity in relation to microbial biomass levels are still lacking. However, we can find observations from other studies that are in line with the constrained enzyme hypothesis."
In the paper intent to provide a possible explanation for some of the observed patterns (such as persistence of small concentrations of substrates at small biomass, but mineralization of those at higher microbial biomass) that are complementary to the explanations given so far.
We extended section 4.2 on the constrained enzyme hypothesis to better set it into context.

RC2-2 Figure 6 illustrates the spatial separation of substrates and enzymes at low biomass levels in the soil (right part) rather than the production of fewer enzyme types.

AC2-2: We did not encounter this unintended interpretation of Figure 6 before. Note that in the right part one type of enzymes is completely absent. Currently, we have no idea of better visualizing the

hypothesis and only clarified the intended meaning in an updated figure caption (now Fig. 9): "There are some substrates for which no enzymes are synthesized".

RC2-3 However, these remarks do not imply that the work is of low quality or incomplete. The idea of allocating resources to the production of different enzyme groups as a response of the microbial community to specific limitation by one or two key nutrients is very valuable and timely. The current MS provides an excellent basis for further solutions in modelling soil microbial biomass and SOM dynamics. The next step would be to test the new model version against experimental data, where the current division into enzyme groups could be confirmed or rejected with suggestions for further model structure modifications.

AC2-3: Thank you for this positive feedback. We are indeed working on a publication where we test the model by fitting it to data from sites of a bedrock phosphorus gradient.

Specific comments and technical notes
RC2-4: Section 2.1: I would suggest providing a model scheme for readers who are facing the SESAM model for the first time. In order not to exactly repeat the scheme of Wutzler et al. 2022, it would be desirable to show the parts (or application points of allocation optimisation approaches) newly developed in this study. It is also not clear from which pool P is obtained. In the 2022 paper there are two enzyme groups for P and in your current publication there seems to be 1, or?

AC2-4: We extended the introduction of the SESAM model and provided the model scheme (section 2.1). Since, this is a cumulative model description series we will repeat only the most relevant parts of the previous model description paper. The version 3.0 described in Wutzler et al. 2022 had no P cycle. However, we derived and compared the optimality approaches already for a P-enabled upcoming version. The two depolymerizing enzymes release P in stoichiometric quantities of both, the L and R source pools (d_ZP at line 93 and eq. 2). However, there is an additional class of P-biomineralizing enzymes that is conceptually different from the depolymerizing enzymes. These enzymes cleave P from the substrates without making C and N available in substrate-stoichiometric quantities at the same time. Therefore, it was important to develop the equations already for this extended case. This is now all described in section 2.1.

RC2-5: L150: There is a misprint, please correct. Seasonal does not seem to be the right term to describe this simulation experiment. Does the temperature and the moisture vary in this case? If not, perhaps 6-month incubation would be a more accurate name for this part.

AC2-5: The simulation experiment did not take into account temperature and moisture effects. This corresponds to holding model drivers, i.e. inputs that can change with time, other than litter inputs constant. In the earth-system modellers community the focus scale of this experiment, which is longer than weekly and shorter than annually, is called "seasonal". Nevertheless, we changed it to "sub-annual" in the revised version (L 227).

RC2-6: L151: Please edit the sentence, as it is not fully clear.

AC2-6: We split the sentence (L 228): "The Incubation simulation experiment added a portion of labile litter to a previously labile-depleted soil. Next, it tracked the carbon use efficiency (CUE) of the microbial community over time and across different C:N ratios of the added labile organic matter."

RC2-7: Fig.2. The legend on the figure can be expanded to make it easier to understand. The meaning of R, L and P can be deduced from the method section, where the numerical experiment is described, but a direct description, i.e. residues, labile and phosphorus-targeted enzymes, would improve the figure.

AC2-7: Thanks for noting. We extended the figure caption of now Fig. 4.

---

## Author Response (AR2)

We thank Stefano Manzoni for making us aware of the typos and language problems. We agree and took care of all his comments (see diff-version of the revised manuscript at the corresponding line-numbers).

RC1-1: Throughout: I would suggest adding the word 'litter' or 'OM' or 'compartment' after 'labile', as it is an adjective (e.g., L228 and Figure 4—both x-axis label and caption)

AC1-1: We consistently used "labile substrate" in the revised manuscript.

L27: "(CUE) is key…"

L33: "knowledge of microbial…"

L54: perhaps I'd delete "recently" (repeated a few words later)

L76: "developments… their own paper"

Table 3: please add 'tvr' in the table

L138: "enzymes with lowest…"

L243: I would remind the reader that this comparison is meaningful because the half saturation constant here had dimensions of a flux (as opposed to the more common dimensions of a stock or content)

L454: delete "several"?

L458: "to simplify…"